# Ecological feedback in quorum-sensing microbial populations can induce heterogeneous production of autoinducers

Matthias Bauer[1†‡§], Johannes Knebel[1†], Matthias Lechner[1], Peter Pickl[2], Erwin Frey[1*]

[1]Arnold Sommerfeld Center for Theoretical Physics and Center for NanoScience, Department of Physics, Ludwig-Maximilians-Universität München, München, Germany; [2]Department of Mathematics, Ludwig-Maximilians-Universität München, München, Germany

*For correspondence: frey@lmu.de

[†]These authors contributed equally to this work

Present address: [‡]Max Planck Institute for Intelligent Systems, Tübingen, Germany; [§]Department of Engineering, University of Cambridge, Cambridge, United Kingdom

Competing interests: The authors declare that no competing interests exist.

**Abstract** Autoinducers are small signaling molecules that mediate intercellular communication in microbial populations and trigger coordinated gene expression via 'quorum sensing'. Elucidating the mechanisms that control autoinducer production is, thus, pertinent to understanding collective microbial behavior, such as virulence and bioluminescence. Recent experiments have shown a heterogeneous promoter activity of autoinducer synthase genes, suggesting that some of the isogenic cells in a population might produce autoinducers, whereas others might not. However, the mechanism underlying this phenotypic heterogeneity in quorum-sensing microbial populations has remained elusive. In our theoretical model, cells synthesize and secrete autoinducers into the environment, up-regulate their production in this self-shaped environment, and non-producers replicate faster than producers. We show that the coupling between ecological and population dynamics through quorum sensing can induce phenotypic heterogeneity in microbial populations, suggesting an alternative mechanism to stochastic gene expression in bistable gene regulatory circuits.

## Introduction

Autoinducers are small molecules that are produced by microbes, secreted into the environment, and sensed by the cells in the population (*Keller and Surette, 2006*; *Hense and Schuster, 2015*). Autoinducers can trigger a collective behavior of all cells in a population, which is called quorum sensing. For example, quorum sensing regulates the transcription of virulence genes in the Gram-positive bacterium *Listeria monocytogenes* (*Gray et al., 2006*; *Garmyn et al., 2011*; *da Silva and De Martinis, 2013*) and the transcription of bioluminescence genes in the Gram-negative bacterium *Vibrio harveyi* (*Xavier and Bassler, 2003*; *Anetzberger et al., 2009*), and it may also autoregulate the transcription of autoinducer synthase genes (*Fuqua and Greenberg, 2002*; *Waters and Bassler, 2005*). When the concentration of autoinducers reaches a threshold value, a coordinated and homogeneous expression of target genes may be initiated in all cells of the population (*Waters and Bassler, 2005*; *Hense and Schuster, 2015*; *Papenfort and Bassler, 2016*), or a heterogeneous gene expression in the population may be triggered at low concentrations (*Anetzberger et al., 2009*; *Williams et al., 2008*; *Boedicker et al., 2009*; *Garmyn et al., 2011*; *Pérez and Hagen, 2010*; *Ackermann, 2015*; *Grote et al., 2014*, *Grote et al., 2015*; *Papenfort and Bassler, 2016*;

**eLife digest** Bacteria and other microbes can communicate with each other using chemical languages. They release small signaling molecules called autoinducers into their surroundings and sense the levels of the autoinducers in the environment. The response to these autoinducers – known as quorum sensing – can regulate how whole communities of microbes grow and behave; for example, autoinducers can alter the ability of microbes to infect humans or enable the microbes to collectively switch on light production.

Recent experiments suggest that, in a population of genetically identical microbes, some individuals may produce autoinducers while others do not. The coexistence of these different "phenotypes" in one population may enable different individuals to perform different roles, or act as a "bet-hedging" strategy that helps the population to survive if it is later exposed to a stressful situation.

It is not clear how microbes regulate autoinducer production so that only some individuals produce these molecules. Bauer, Knebel et al. developed a theoretical model to address this question. In the model, the microbes shape their environment by producing autoinducers and can respond to this self-shaped environment by changing their level of autoinducer production. Bauer, Knebel et al. found that this establishes a feedback loop that can result in autoinducers being produced by some individuals but not others.

The next step following on from this work is to carry out experiments to test the assumptions and predictions made by the theoretical model. These findings may help to understand how the coexistence of different phenotypes affects collective behaviors, and vice versa, in populations of microbes that use quorum-sensing.

*Pradhan and Chatterjee, 2014*). To implement all of these functions and behaviors, a microbial population needs to dynamically self-regulate the average autoinducer production.

Within a given population, the promoter activity of autoinducer synthase genes may vary between genetically identical cells (*Garmyn et al., 2011*; *Grote et al., 2014*; *Anetzberger et al., 2012*; *Plener et al., 2015*; *Cárcamo-Oyarce et al., 2015*; *Grote et al., 2015*). For example, during the growth of *L. monocytogenes* under well-mixed conditions two subpopulations were observed, one of which expressed autoinducer synthase genes, while the other did not (*Garmyn et al., 2011*). Such a phenotypic heterogeneity was associated with biofilm formation (*Garmyn et al., 2011*; *da Silva and De Martinis, 2013*; *Hense and Schuster, 2015*; *Cárcamo-Oyarce et al., 2015*). The stable coexistence of different phenotypes in one population may serve the division of labor or act as a bet-hedging strategy and, thus, may be beneficial for the survival and resilience of a microbial species on long time scales (*Ackermann, 2015*).

The mechanism by which a heterogeneous expression of autoinducer synthase genes is established when their expression is autoregulated by quorum sensing has remained elusive. For example, expression of the above mentioned autoinducer synthase genes in *L. monocytogenes* is up-regulated through quorum-sensing in single cells (*Garmyn et al., 2011*, *Garmyn et al., 2009*; *Waters and Bassler, 2005*). From an experimental point of view it is often not known, however, whether autoinducer synthesis is up-regulated for all autoinducer levels or only above a threshold level. To explain phenotypic heterogeneity of autoinducer production, currently favored threshold models of quorum sensing typically assume a bistable gene regulation function (*Fujimoto and Sawai, 2013*; *Pérez-Velázquez et al., 2016*; *Goryachev et al., 2005*; *Dockery and Keener, 2001*). For bistable regulation, cellular autoinducer synthesis is up-regulated above a threshold value of the autoinducer concentration in the population, whereas it is down-regulated below the threshold ('all-or-none' expression); see *Figure 1B*. Stochastic gene expression at the cellular level then explains the coexistence of different phenotypes in one population. If, however, cellular autoinducer synthesis is up-regulated for all autoinducer concentrations (monostable up-regulation), the mechanism by which phenotypic heterogeneity can arise and is controlled has not been explained.

Here we show that the coupling between ecological and population dynamics through quorum sensing can control a heterogeneous production of autoinducers in quorum-sensing microbial

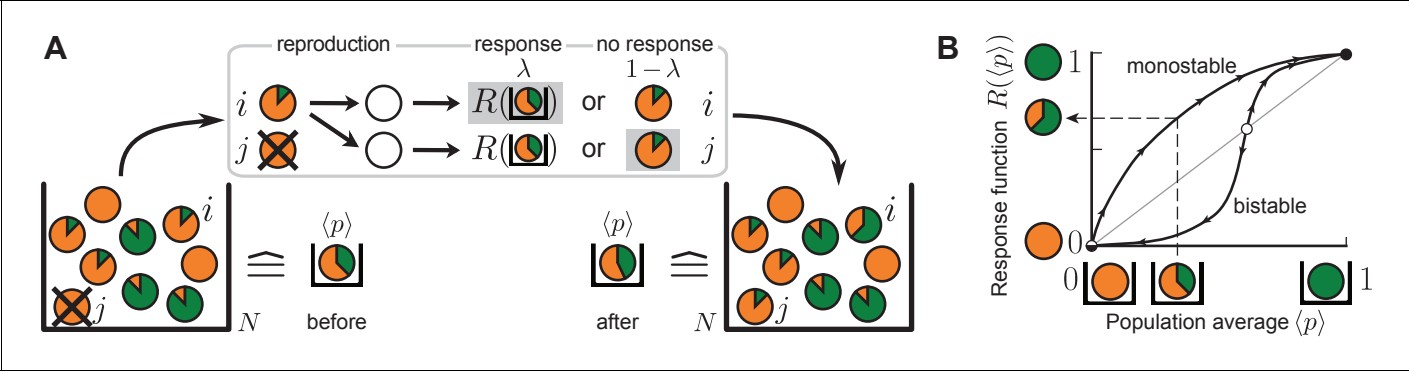

**Figure 1.** The quorum-sensing model for the production of autoinducers in microbial populations. (**A**) Sketch of a typical update step. Individuals are depicted as disks and the degree of autoinducer production ($p_i \in [0, 1]$) is indicated by the size of the green fraction. Non-producers (orange disks) reproduce fastest, full producers (green disks) slowest. Individual $i$ with $p_i = 1/6$ divides into two offspring individuals, one of which replaces another individual $j$. Both offspring individuals sense the average production level in the population ($\langle p \rangle = 1/3$), and may either respond to this environment, with probability $\lambda$, by adopting the value $R(\langle p \rangle)$ of the response function ($= 2/3$ here, see (**B**)) or, with probability $1 - \lambda$, retain the production degree from the ancestor ($= 1/6$). Here, offspring individual $i$ responds to the environment while $j$ does not (denoted by gray shading). (**B**) Quorum sensing is characterized by the response function. Perception of the average production level in the population ($\langle p \rangle$) enables individuals to change their production degree to the value $R(\langle p \rangle) \in [0, 1]$. Sketched are a monostable response function (stable fixed point at 1, unstable fixed point at 0), and a bistable response function (stable fixed points at 0 and 1, unstable fixed point at an intermediate threshold value). Stable fixed points of the response function are depicted as black circles while unstable fixed points are colored in white. For the sketched bistable response function, autoinducer production is down-regulated with respect to the sensed production level in the population below the threshold value, and up-regulated above this threshold. For the monostable response function, autoinducer production is up-regulated at all sensed production levels.

populations. At the same time, the overall autoinducer level in the environment is robustly self-regulated, so that further quorum-sensing functions such as virulence or bioluminescence can be triggered. We studied the collective behavior of a stochastic many-particle model of quorum sensing, in which cells produce autoinducers to different degrees and secrete them into the well-mixed environment. Production of large autoinducer molecules (for example oligopeptides) and accompanied gene expression are assumed to reduce fitness such that non-producers reproduce faster than producing cells. Moreover, it is assumed that quorum sensing enables up-regulation of autoinducer production, that is, individuals can increase their production in response to the sensed average production level in the population (*Figure 1*). As a central result, we found that the population may split into two subpopulations: one with a low, and a second with a high production rate of autoinducers. This phenotypic heterogeneity in the autoinducer production is stable for many generations and the autoinducer concentration in the population is tightly controlled by how production is up-regulated. If cellular response to the environment is absent or too frequent, phase transitions occur from heterogeneous to homogeneous populations in which all individuals produce autoinducers to the same degree. To capture these emergent dynamics, we derived the macroscopic mean-field equation *(1)* from the microscopic stochastic many-particle process in the spirit of the kinetic theory in statistical physics, which we refer to as the *autoinducer equation*. The analysis of the autoinducer equation explains both phenotypic heterogeneity through quorum sensing and the phase transitions to homogeneity.

The key aspect of our work is how the composition of a population changes in time when its constituents respond to an environment that is being shaped by their own activities (see *Box 1*). This ecological feedback is mediated by quorum sensing and creates an effective global coupling between the individuals in the population. Such a global coupling is reminiscent of long-range interactions in models of statistical mechanics, such as in the classical XY spin model with infinite range interactions (*Antoni and Ruffo, 1995*; *Yamaguchi et al., 2004*; *Barré et al., 2002*; *Choi and Choi, 2003*; *de Buyl et al., 2010*; *Campa et al., 2009*; *Pakter and Levin, 2013*). Our analysis suggests that quorum sensing in microbial populations can induce and control phenotypic heterogeneity as a collective behavior through such a global coupling and, notably, does not rely on a bistable gene regulatory circuit (see *Discussion*).

## Set-up of the quorum-sensing model

We now introduce the quorum-sensing model for a well-mixed population of $N$ individuals (*Figure 1*). The phenotype of each individual $i = 1, \ldots, N$ is characterized by its production degree $p_i \in [0, 1]$, that is, the extent to which it produces and secretes autoinducers. In an experiment with microbes, the promoter activity of autoinducer synthase genes or their enzymatic activity could be a proxy for the production degree. The limiting case $p_i = 0$ denotes a non-producer, and $p_i = 1$ denotes a full producer.

The state of the population $\mathbf{p} = (p_1, \ldots, p_N)$ changes stochastically (*Figure 1A*): An individual $i$ reproduces with rate $\phi_i$, which we refer to as the individual's fitness. We assume that fitness decreases with incurring metabolic costs of induction and synthesis of autoinducers, and with other metabolic burdens in the cell's phenotypic state (*Ruparell et al., 2016*; *Diggle et al., 2007*; *He et al., 2003*). For simplicity, we choose $\phi_i = \phi(p_i) = 1 - s p_i$. The selection strength $0 \leq s < 1$ scales the fitness difference with respect to the non-producing phenotype ($\phi(0) = 1$). Thus, the larger an individual's production, the smaller its reproduction rate. This assumption is discussed in detail further below (see *Discussion*).

Whenever an individual divides into two offspring individuals in the stochastic process, another individual from the population is selected at random to die such that the population size $N$ remains constant. Qualitative results of our model remain valid if only the average population size is constant, which may be assumed, for example, for the stationary phase of microbial growth in batch culture. One recovers the mathematical set-up of frequency-dependent Moran models for Darwinian selection (*Moran, 1958*; *Ewens, 2004*; *Blythe and McKane, 2007*; *Nowak et al., 2004*) if one restricts the production degrees to a discrete set, for example, to full producers or non-producers only, $p_i \in \{0, 1\}$. The mathematical set-up of the well-known Prisoner's dilemma in evolutionary game theory is recovered if, in addition, the secreted molecules would confer a fitness benefit on the population (*Nowak et al., 2004*; *Traulsen et al., 2005*; *Melbinger et al., 2010*; *Assaf et al., 2013*). Since we are interested in the mechanism by which heterogeneous production of autoinducers might be induced and do not study the context under which it might have evolved, we do not include any fitness benefits through signaling, for example at the population level, into the modeling here (see *Discussion*).

A central feature of our model is the fact that individuals may adjust their production degree via a sense-and-response mechanism through quorum sensing, which is implemented as follows. After reproduction, both offspring individuals sense the average production level of autoinducers $\langle p \rangle = 1/N \sum_i p_i$ in the well-mixed population. With probability $\lambda$, they independently adopt the value $R(\langle p \rangle) \in [0, 1]$ as their production degree in response to the sensed environmental cue $\langle p \rangle$, whereas they retain the ancestor's production degree with probability $1 - \lambda$ through non-genetic inheritance. In an experimental setting, the response probability $\lambda$ relates to the rate with which cells respond to the environment (*Kussell and Leibler, 2005*; *Acar et al., 2008*; *Axelrod et al., 2015*) and regulate their production through quorum sensing. We refer to the function $R(\langle p \rangle)$ as the response function, which is the same for all individuals. The response function encapsulates all biochemical steps involved in the autoinducer production between perception of the average production level $\langle p \rangle$ and adjustment of the individual production degree to $R(\langle p \rangle)$ in response (*He et al., 2003*; *Williams et al., 2008*; *Drees et al., 2014*; *Hense and Schuster, 2015*; *Maire and Youk, 2015*); see *Figure 1B*. For example, it may be a bistable step or bistable Hill function, which is often effectively assumed in threshold models of phenotypic heterogeneity (*Fujimoto and Sawai, 2013*; *Pérez-Velázquez et al., 2016*; *Goryachev et al., 2005*; *Dockery and Keener, 2001*). For a bistable response function, cellular production is up-regulated above a threshold value of $\langle p \rangle$, whereas it is down-regulated below the threshold. For the bistable response function sketched in *Figure 1B*, both values $\langle p \rangle = 0$ and $\langle p \rangle = 1$ are stable fixed points. In this work, however, we particularly focus on monostable response functions $R(\langle p \rangle)$ to model microbial quorum-sensing systems in which autoinducer synthesis is up-regulated at all autoinducer production levels in the population (*Garmyn et al., 2009*; *Waters and Bassler, 2005*). In other words, cellular production always increases with respect to the sensed production level in the population (stable fixed point at $\langle p \rangle = 1$ and unstable fixed point at $\langle p \rangle = 0$). The sense-and-response mechanism is further discussed in the *Discussion* section.

From a mathematical point of view, the introduced sense-and-response mechanism through quorum sensing constitutes a source of innovation in the space of production degrees because an individual may adopt a production degree that was not previously present in the population. Thus, a continuous production space with $p_i \in [0, 1]$ as opposed to a discrete production space is a technical

necessity for the implementation of the quorum-sensing model. The coupling of ecological dynamics (given by the average production level of autoinducers $\langle p \rangle$) with population dynamics (determined by fitness differences between the phenotypes) through quorum sensing results in interesting collective behavior, as we show next. We emphasize that, as long as this coupling is present, the effects of the quorum-sensing model that we found and report next are qualitatively robust against noise at all steps; see below.

## Box 1. An ecological feedback can control phenotypic heterogeneity in quorum-sensing microbial populations.

Our work demonstrates that the coupling of ecological and population dynamics through quorum sensing cannot only lead to homogeneously producing populations, but can also control a heterogeneous production of autoinducers in microbial populations. Phenotypic heterogeneity becomes manifest in the quorum-sensing model as long-lived, bimodal states of the population that are dynamically stable; see sketch below and *Equation (2)*.

In the quorum-sensing model, ecological dynamics are determined by the average production level of autoinducers, while population dynamical changes are determined by fitness differences between non-producers and producers of autoinducers. Because individuals sense and respond to autoinducers in the environment, the ecological dynamics are coupled with the population dynamics. In other words, an ecological feedback loop is established when cells respond to an environment that is being shaped by their own activities. When fitness differences between non-producers and producers of autoinducers balance with cellular response to autoinducers in the environment, *separated* production degrees stably coexist in one population. Therefore, we expect that a heterogeneous production of autoinducers may be induced and controlled by such an ecological feedback in real microbial populations, suggesting an alternative mechanism to stochastic gene expression in bistable gene-regulatory circuits to control phenotypic heterogeneity (see *Discussion*).

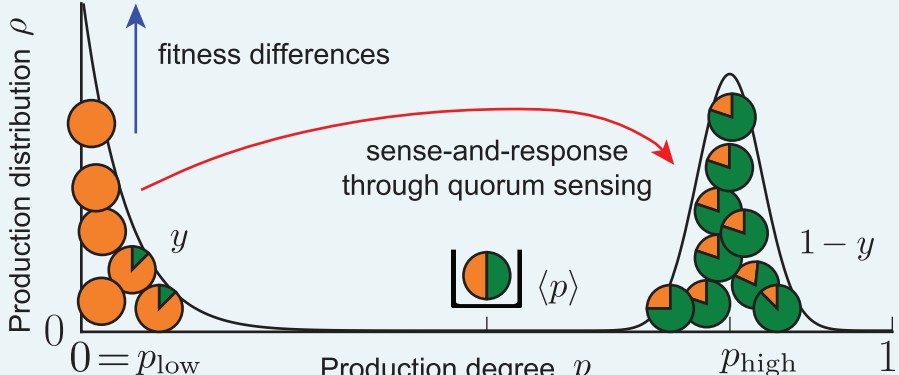

**Box 1—figure 1.** Effective picture of robust phenotypic heterogeneity through an ecological feedback. The coupling of fitness differences between non-producers and producers (selection strength *s*) and sense-and-response to the self-shaped environment through quorum sensing (response probability $\lambda$ and up-regulation of production with response function $R(\langle p \rangle)$) ensures the stable coexistence of the two subpopulations at the phenotypic states $p_{\text{low}}$ and $p_{\text{high}}$; see *Equation (2)*. The value $\beta = 2\lambda/s$ quantifies this coexistence. In one subpopulation (fraction $y = 1 - \beta/R(\beta)$ of the total population), individuals do not produce ($p_{\text{low}} = 0$), while in the other (fraction $1 - y$) individuals produce autoinducers to the degree $p_{\text{high}} = R(\beta)$. The average production level in the population is robustly adjusted to the value $\langle p \rangle = \beta$. States of phenotypic heterogeneity arise for a broad range of initial distributions and are robust against noisy inheritance, noisy perception, and noisy response (see *Results of mathematical analysis* and *Appendix 1—figures 1* and *2*).

## Results of numerical simulations

The quorum-sensing model was numerically simulated by employing Gillespie's stochastic simulation algorithm (*Gillespie, 1976, 1977*) for a population size of $N = 10^4$ individuals and an exemplary selection strength $s = 0.2$, such that $sN \gg 1$. In this regime, demographic fluctuations are subordinate (*Nowak et al., 2004*; *Wild and Traulsen, 2007*; *Blythe and McKane, 2007*). Within the scope of our quorum-sensing model, the precise value of the selection strength $s$ that scales the fitness differences is not important for the reported mechanism by which phenotypic heterogeneity can be induced, see below. We tracked the state of the population **p** over time, and depict the histogram of production degrees and the population average in *Figure 2*.

First, we studied the stochastic many-particle process without sense-and-response ($\lambda = 0$); see *Figure 2A,D* and *Video 1*. In this case, non-producers always proliferate because they reproduce at the highest rate in the population, which is well-studied in evolutionary game theory (*Taylor and*

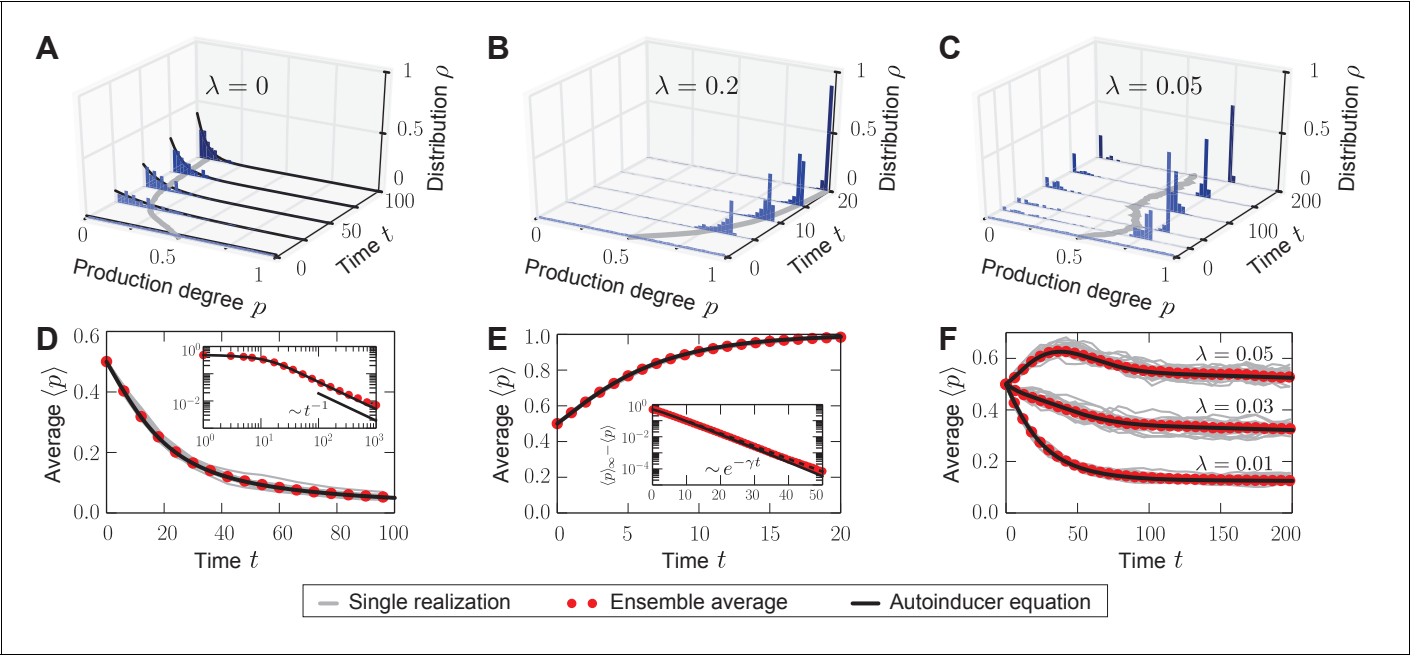

**Figure 2.** Homogeneous and heterogeneous production of autoinducers in the quorum-sensing model. Temporal evolution of autoinducer production in the quorum-sensing model depicted as histograms of production degrees (normalized values), (A–C); and average production level of autoinducers in the population (D–F); see also *Videos 1–3*. (A) In the absence of sense-and-response ($\lambda = 0$), only non-producers proliferate. The approach to stationarity is asymptotically algebraically slow for a quasi-continuous initial distribution of production degrees (D). The black line $\langle p \rangle \sim t^{-1}$ serves as a guide for the eye. (B) Sense-and-response through quorum sensing ($\lambda = 0.2$ here) promotes autoinducer production, and the population becomes homogeneous (ultimately, fixation at a single production degree, data not shown). The response function used here, $R(\langle p \rangle) = \langle p \rangle + 0.2 \cdot \sin(\pi \langle p \rangle)$, was chosen such that an individual's production degree is always up-regulated through quorum sensing (see *Figure 1B*). Approach to stationarity is exponentially fast (E), but timescales may diverge at bifurcations of the response function (see *Appendix 1—figure 3*). The dashed line in (E) shows fit to an exponential decay. (C) When $\lambda$ is small ($\lambda = 0.05$ here), the population becomes heterogeneous: quasi-stationary states arise in which the population splits into two subpopulations, one of which does not produce autoinducers, while the other does. The same monostable response function was chosen as in (B). Therefore, heterogeneity may arise without bistable response. For very long times, one of the two absorbing states (A, B) is reached, data not shown (see *Figure 3A*). Heterogeneous, quasi-stationary states arise for a broad class of initial distributions (see *Appendix 1—figure 1* and our mathematical analysis). At the same time, the average production level of autoinducers in the population is adjusted by the response probability $\lambda$ if $s$ is fixed (F) or vice versa (data not shown). Bimodal, quasi-stationary states also arise when noisy inheritance, noisy perception, and noisy response are included in the model set-up (see *Appendix 1—figure 2*). Mean-field theory agrees with all observations (autoinducer equation (1)). The time unit $\Delta t = 1$ means that in a population consisting solely of non-producers, each individual will have reproduced once on average. Ensemble size $M = 100$, $s = 0.2$, $N = 10^4$.

The following source data is available for figure 2:

**Source data 1.** Source data accompanying *Figure 2*.

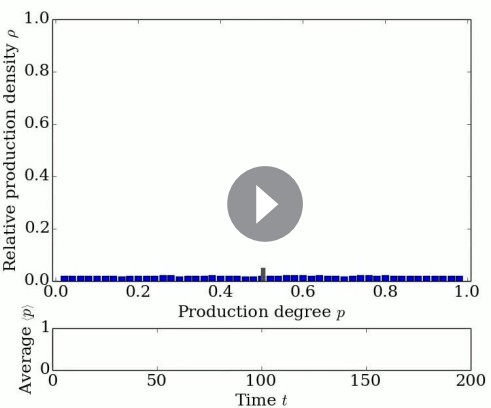

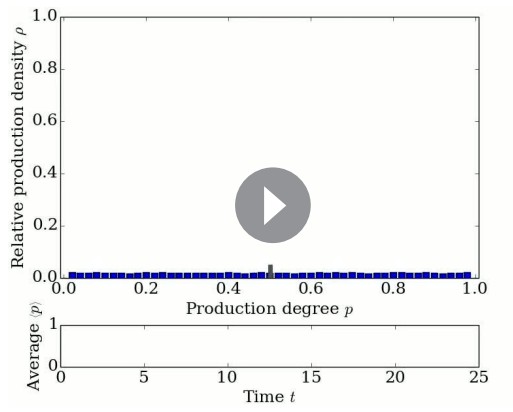

**Video 1.** Video accompanying *Figure 2A* – Homogeneous production of autoinducers in the population if sense-and-response is absent in the quorum-sensing model ($\lambda = 0$).

**Video 2.** Video accompanying *Figure 2B* – Homogeneous production of autoinducers in the population if sense-and-response is frequent in the quorum-sensing model ($\lambda = 0.2$ here).

*Jonker, 1978*; *Maynard Smith, 1982*; *Hofbauer and Sigmund, 1998*). Thus, the initially uniform distribution in the population shifts to a peaked distribution at low production degrees. Ultimately, a *homogeneous* (unimodal) stationary state is reached in which all individuals produce autoinducers to the same low degree $p_{\mathrm{low}} \simeq 0$. Such a stationary state is absorbing (*Hinrichsen, 2000*), that is, the stochastic process offers no possibility of escape from this state of the population.

With quorum sensing ($\lambda > 0$), absorbing states are reached if, again, all individuals produce to the same degree $p^*$ and, in addition, the value of this production degree is a fixed point of the response function ($R(p^*) = p^*$); see *Figure 2B,E* and *Video 2*. In such a homogeneous absorbing state with $\langle p \rangle_\infty = p^*$, an offspring individual can no longer alter its production degree. It either takes over the production degree $p^*$ from its ancestor or it adopts that same degree $R(\langle p \rangle_\infty) = \langle p \rangle_\infty = p^*$ through sense-and-response. Thus, all individuals continue to produce with degree $p^*$ and the state of the population remains *homogeneous* (unimodal).

Surprisingly, for small response probabilities $\lambda$, we found that the population may get trapped in *heterogeneous* (bimodal) states for long times before a homogeneous absorbing state is reached. The temporal evolution of such a heterogeneous state is shown in *Figure 2C,F* and *Video 3* for $\lambda = 0.05$. A monostable response function was chosen with $R(\langle p \rangle) > \langle p \rangle$ for all $\langle p \rangle \in (0,1)$ (unstable fixed point at 0, and stable fixed point at 1) such that the production degree is always up-regulated through quorum sensing; see sketch in *Figure 1B*. After some time has elapsed, the population is composed of two subpopulations: one in which individuals produce autoinducers to a low degree $p_{\mathrm{low}}$, and a second in which individuals produce to a higher degree $p_{\mathrm{high}}$ that is separated from $p_{\mathrm{low}}$ by a gap in the space of production degrees. Only through strong demographic fluctuations can the population reach one of the homogeneous absorbing states ($\langle p \rangle_\infty = 0$ or 1 for the response function chosen above). The time taken to reach a homogeneous absorbing state grows exponentially with $N$ (*Figure 3A*). Therefore, states of phenotypic heterogeneity are quasi-stationary and long-lived. These heterogeneous states arise for a broad class of response

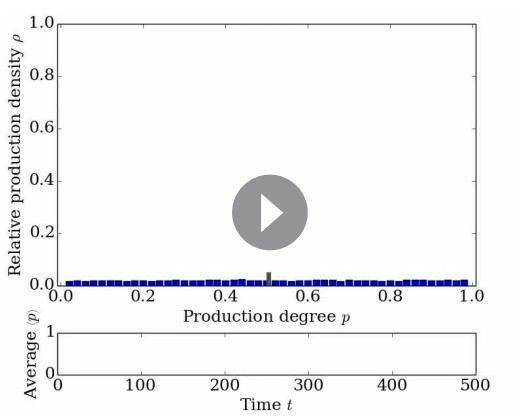

**Video 3.** Video accompanying *Figure 2C* – Heterogeneous production of autoinducers in the population if sense-and-response is rare in the quorum-sensing model ($\lambda = 0.05$ here).

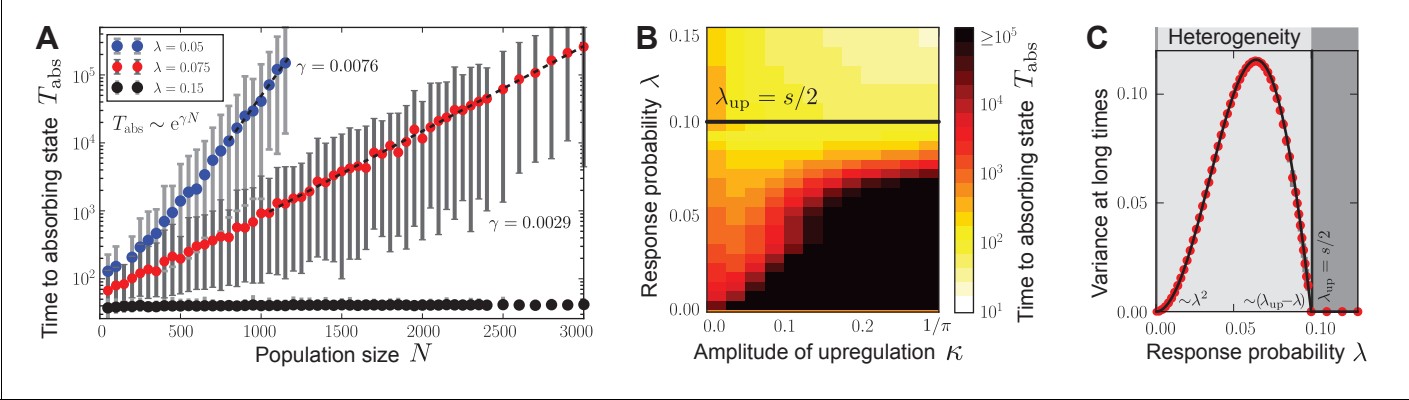

**Figure 3.** Characterization of phenotypic heterogeneity in the quorum-sensing model. (A) For small response probability $\lambda$, populations get stuck in heterogeneous quasi-stationary states. The time taken to reach a homogeneous absorbing state, $T_{abs}$, increases exponentially with the population size $N$ (filled circles denote the mean, gray bars denote the range within which 95% of the data points lie closest to the mean; dashed lines show fit to $T_{abs} \sim e^{\gamma N}$). (B) Heterogeneous states are long-lived only if $\lambda$ is small and the response function is nonlinear (in particular, up-regulation is required for some average production level such that $R(\langle p \rangle) > \langle p \rangle$). Here, the monostable response function $R(\langle p \rangle) = \langle p \rangle + \kappa \sin(\pi \langle p \rangle)$ was chosen such that $\kappa \in [0, 1/\pi]$ scales the magnitude of up-regulation. As $\kappa$ increases, the gap between the low-productive and high-productive peaks of the heterogeneous state becomes larger such that it takes longer to reach the absorbing state. Mean-field theory (1) predicts the existence and local stability of heterogeneous stationary distributions for $0 < \lambda < \lambda_{up} = s/2$ (regime below the black line). Deviations between the stochastic process and mean-field theory are due to demographic fluctuations that vanish as $N \to \infty$. (C) The variance of production degrees in the population reveals whether the population is in a homogeneous ($\mathrm{Var}(p) = 0$) or heterogeneous state ($\mathrm{Var}(p) > 0$). The variance was averaged over long times in the quasi-stationary state. Mean-field theory (1) (black line) agrees with our numerical observations (red filled circles); see *Methods and materials*. Ensemble size $M = 100$, $s = 0.2$, in (B) $N = 10^3$ and in (C) $N = 10^4$ and $N = 5 \cdot 10^4$ close to $\lambda_{up}$, in (A, C) $\kappa = 0.2$.

The following source data is available for figure 3:

**Source data 2.** Source data accompanying *Figure 3*.

functions and initial distributions (*Appendix 1—figure 1*), and they are robust against demographic noise that is always present in populations of finite size (*Figure 3A*); see our mathematical analysis below. We demonstrated that states of phenotypic heterogeneity are also robust against changes of the model set-up, which might account for more biological details (see, for example, *Papenfort and Bassler, 2016* and references therein). Upon including, for example, noisy inheritance of the production degree, noisy perception of the environment, and noisy response to the environment into the quorum-sensing model, heterogeneous states still arise; see *Appendix 1—figure 2*. Furthermore, the average production in the heterogeneous state is finely adjusted by the interplay between the response probability $\lambda$ and the selection strength $s$ (*Figure 2F*).

The establishment of long-lived, heterogeneous states induced by quorum sensing is one central finding of our study. We interpret this phenotypic heterogeneity as the result of the robust balance between population and ecological dynamics coupled through quorum sensing (see *Box 1*). On the one hand, fitness differences due to costly production favor non-producers. On the other hand, sensing the population average and accordingly up-regulating individual production enables producers to persist. Remarkably, fitness differences and sense-and-response balance such that *separated* production degrees may stably coexist in one population; the population does not become homogeneous at an intermediate production degree as one might naively expect. Heterogeneity of the autoinducer production is a robust outcome of the dynamics (and not a fine-tuned effect), and the average production level in the population is adjusted by the interplay of the response probability $\lambda$ and the selection strength $s$. Phenotypic heterogeneity does not rely on a bistable response function, but arises due to the global intercellular coupling of ecological and population dynamics through quorum sensing, as we show next. The relevance of quorum sensing for phenotypic heterogeneity in microbial populations is further explored below (see *Discussion*).

## Results of mathematical analysis

In the following, the observed long-lived states of phenotypic heterogeneity in the quorum-sensing model are explained. First, we derived the macroscopic mean-field equation (the *autoinducer equation (1)*) from the microscopic dynamics of the quorum-sensing model. Second, we analyzed this mean-field equation and characterized phenotypic heterogeneity of autoinducer production.

The microscopic dynamics of the quorum-sensing model are captured by a memoryless stochastic birth-death process as sketched in *Figure 1*. Starting from the microscopic many-particle stochastic process, we derived a mean-field equation for the probability distribution of finding *any* individual at a specified production degree $p$ at time $t$ in the spirit of the kinetic theory in statistical physics (*Kadar, 2007*). We call this one-particle probability distribution the production distribution $\rho$; *Figure 2* shows the corresponding histogram numerically obtained from the stochastic many-particle process. The mean-field equation for $\rho$, which we refer to as the *autoinducer equation*, is obtained as:

$$\partial_t \rho(p,t) = 2\lambda \overline{\phi}_t \big( \delta(p - R(\overline{p}_t)) - \rho(p,t) \big) + (1 - 2\lambda)\big( \phi(p) - \overline{\phi}_t \big) \rho(p,t) , \tag{1}$$

where $\overline{\cdot}_t$ denotes averaging with respect to $\rho$ at time $t$. The details of the derivation of the autoinducer equation from the microscopic dynamics are given in the *Methods and materials* section and in *Appendix 2*.

The autoinducer equation *(1)* involves two contributions: the *sense-and-response* term with prefactor $2\lambda$, and the *replicator* term with prefactor $1 - 2\lambda$. Through the replicator term, probability weight at production degree $p$ changes if the fitness $\phi(p)$ is different from the mean fitness in the population $\overline{\phi}_t$ (here $\phi(p) - \overline{\phi}_t = -s(p - \overline{p}_t)$). Without quorum sensing ($\lambda = 0$), *Equation (1)* reduces to the well-known replicator equation of the continuous Prisoner's dilemma (*Bomze, 1990*; *Oechssler and Riedel, 2001*; *Hofbauer and Sigmund, 2003*; *Cressman, 2005*; *McGill and Brown, 2007*). The sense-and-response term, on the other hand, encodes the global feedback by which individuals adopt the production degree $R(\overline{p}_t)$ upon sensing the average $\overline{p}_t$ through quorum sensing at rate $2\lambda$. The difference between the current state $\rho$ and the state in which all individuals have this production degree $R(\overline{p}_t)$ determines the change in $\rho$ at every production degree. Through the replicator term and the sense-and-response term, the ecological dynamics (average production level $\overline{p}_t$) are coupled with the dynamics of $\rho$.

We now present our results for the long-time behavior of the autoinducer equation *(1)*. First, the autoinducer equation *(1)* admits homogeneous stationary distributions. Without quorum sensing ($\lambda = 0$), the initially lowest production degree in the population, $p_{\text{low}}$, constitutes the *homogeneous* stationary distribution $\rho_\infty(p) = \delta(p - p_{\text{low}})$, which is attractive for generic initial conditions. With quorum sensing ($\lambda > 0$), fixed points of the response function $p^* = R(p^*)$ yield *homogeneous* stationary distributions as $\rho_\infty(p) = \delta(p - p^*)$, which are attractors of the quorum-sensing dynamics *(1)* for all initial distributions if $\lambda > s/2$; see analysis below. These homogeneous stationary distributions confirm our observations of homogeneous absorbing states in the quorum-sensing model, in which all individuals produce to the same degree; see *Figure 2A,B*. Time scales at which stationarity is approached are discussed in the *Methods and materials* section.

Second, to analytically characterize long-lived heterogeneous states of the population, we decomposed $\rho$ into a distribution at low production degrees and a remainder distribution at higher degrees. We found that such a decomposition yields the bimodal, *heterogeneous*, stationary distribution of the autoinducer equation *(1)*:

$$\begin{aligned} \rho_\infty(p) &= y\delta(p) + (1-y)\delta(p - p_{\text{high}}) , \\ \text{with } &p_{\text{high}} = R(\beta) \ \text{and} \ y = 1 - \beta/R(\beta) , \end{aligned} \tag{2}$$

if the conditions $0 < p_{\text{high}} \leq 1$ and $0 < y < 1$ are fulfilled; see *Box 1* for an illustration and *Appendix 3* for the derivation. The parameter $\beta = 2\lambda/s$ quantifies the balance between fitness differences and sense-and-response mechanism through quorum sensing. Heterogeneous stationary distributions *(2)* are constituted of a probability mass $y$ at the low-producing degree $p_{\text{low}} = 0$ and a coexisting $\delta$-peak with stationary value $1 - y$ at a high-producing degree $p_{\text{high}}$ separated from $p_{\text{low}}$ by a gap. Such heterogeneous stationary distributions have mean $\overline{p}_\infty = \beta$ and variance $\text{Var}(p)_\infty = \beta(R(\beta) - \beta)$. Therefore, the interplay between selection strength $s$ and response probability $\lambda$ adjusts the average

production of autoinducers in the population (*Figure 2F*). For simplicity, we assumed in *Equation (2)* that the initially lowest production degree in the population is $p_{low} = 0$; generalized bimodal distributions for arbitrary initial distributions $\rho_0$ are given in *Appendix 3*.

From the conditions on $p_{high}$ and $y$ below *Equation (2)*, one can derive the following conditions on the response function and the value of the response probability $\lambda$ (for given selection strength $s$) for the existence of heterogeneous stationary distributions: (i) The response function needs to be nonlinear with $R(\bar{p}_\infty) = p_{high} > \bar{p}_\infty$; that is, quorum sensing needs to up-regulate the cellular production in some regime of the average production level. Therefore, both monostable and bistable response functions depicted in *Figure 1B* may induce heterogeneous stationary distributions through the ecological feedback. (ii) The response probability needs to be small with $\lambda < \lambda_{up} = s/2$; that is, to induce phenotypic heterogeneity, cells must respond only rarely to the environmental cue $\bar{p}$. This estimate of an upper bound on $\lambda$ is confirmed by our numerical results of the stochastic process (*Figure 3A–C*). Vice versa, for a given response probability, the selection strength needs to be big enough to induce heterogeneous stationary distributions. As we show in the *Methods and materials* section, phase transitions in the space of stationary probability distributions govern the long-time dynamics of the autoinducer equation *(1)* from heterogeneity to homogeneity as the response probability changes ($\lambda \to 0$ and $\lambda \to \lambda_{up}$); see *Figure 3C*.

For small $\lambda$, the coexistence of the low-producing and the high-producing peaks in solution (*2*) is stable due to the balance of fitness differences and sense-and-response through quorum sensing. In *Appendix 3* we show that the heterogeneous stationary distributions (*2*) are stable up to linear order in perturbations around stationarity. As our numerical simulations show, these bimodal distributions are the attractor of the mean-field dynamics *(1)* for a broad range of initial distributions when $\lambda$ is small; see *Appendix 1—figure 1* for some examples. They are also robust against noisy inheritance, noisy perception, and noisy response as demonstrated in *Appendix 1—figure 2*. We interpret the stability of the bimodal stationary distributions (*2*) as follows (see also *Box 1*). Fitness differences quantified by the selection strength $s$ increase probability mass at production degree $p_{low}$, whereas nonlinear response to the environment with probability $\lambda$ pushes probability mass towards the up-regulated production degree $p_{high} = R(\bar{p}_\infty)$. The gap $p_{high} - p_{low} > 0$ ensures that the exponential time scales of selection and sense-and-response stably balance the coexistence of both peaks; see *Methods and materials*. Because heterogeneous stationary distributions (*2*) are attractive and stable, heterogeneous states of the stochastic many-particle process arise and are quasi-stationary. Consequently, the time to reach a homogeneous absorbing state in the stochastic process through demographic fluctuations scales exponentially with the population size $N$ (*Elgart and Kamenev, 2004*; *Kessler and Shnerb, 2007*; *Assaf and Meerson, 2010*; *Frey, 2010*; *Hanggi, 1986*); see *Figure 3A*. Thus, phenotypic heterogeneity is long-lived.

In summary, our mathematical analysis explains how phenotypic heterogeneity in the autoinducer production arises when quorum sensing up-regulates the autoinducer production in microbial populations (*Box 1*). As an emergent phenomenon, the population may split into two subpopulations: one in which cells do not produce autoinducers ('off' state, $p_{low} = 0$) and a second in which cells produce autoinducers ('on' state, $p_{high} = R(2\lambda/s) > 0$), but grow slower. The fraction of individuals in the 'off' state is given by the value of $y$ in *Equation (2)*. If quorum sensing is absent ($\lambda = 0$), the whole population is in the 'off' state ($y = 1$), whereas all individuals are in the 'on' state ($y = 0$) if quorum sensing is frequent ($\lambda \geq \lambda_{up}$). Only when response to the environment is rare ($0 < \lambda < \lambda_{up}$) can the two phenotypic states, $p_{low}$ and $p_{high}$, coexist in the population ($0 < y < 1$). The transitions from heterogeneous to homogeneous populations are governed by nonequilibrium phase transitions when the response probability changes ($\lambda \to 0$ and $\lambda \to \lambda_{up}$). Our mathematical analysis shows that phenotypic heterogeneity arises dynamically, is robust against perturbations of the autoinducer production in the population, and is robust against noise at the level of inheritance, sense, and response.

## Discussion

### Summary: Phenotypic heterogeneity in the quorum-sensing model as a collective phenomenon through an ecological feedback

In this work, we studied a conceptual model for the heterogeneous production of autoinducers in quorum-sensing microbial populations. The two key assumptions of our quorum-sensing model are

as follows. First, production of large autoinducer molecules and accompanied gene expression in the cell's phenotypic state are negatively correlated with fitness such that non-producers reproduce faster than producers. Second, cells sense the average production level of autoinducers in the population and may accordingly up-regulate their production through quorum sensing. As a result, not only does the interplay between fitness differences and sense-and-response give rise to homogeneously producing populations, but it can also induce a heterogeneous production of autoinducers in the population as a stable collective phenomenon. In these heterogeneous states, the average production level of autoinducers in the population is adjusted within narrow limits by the balance between fitness differences (selection strength $s$ in the model), and the rate with which cells respond to the environment and up-regulate their production through quorum sensing (response probability $\lambda$ and response function $R(\langle p \rangle)$ in the model). Due to this robust adjustment of the production level in the population, the expression of other genes (for example, bioluminescence and virulence genes) can be regulated by quorum sensing even when the production of autoinducers is heterogeneous in the population.

In the following, we discuss the assumptions of our model in the light of the empirical reality for both quorum sensing and phenotypic heterogeneity. Furthermore, we indicate possible directions to experimentally test the ecological feedback that is suggested by the results of our theoretical work.

## Does autoinducer production reduce individual growth rate?

In our quorum-sensing model, it is assumed that the individual's production degree of autoinducers is negatively correlated with its growth rate ($\phi_i = 1 - s p_i$). Is this assumption of growth impairment for producing phenotypes justified (*Parsek and Greenberg, 2005*)? This would be the case if cellular production of autoinducers directly causes a reduction of the cell's growth rate. For example, in *L. monocytogenes* populations, heterogeneous production was observed for an autoinducer oligopeptide that is synthesized via the *agr* operon (*Garmyn et al., 2011*, *Garmyn et al., 2009*). This signaling oligopeptide incurs high metabolic costs through the generation of a larger pre-protein. For the oligopeptide signal synthesized via the *agr* operon in *Staphylococcus aureus*, the metabolic costs were conservatively estimated by Keller and Surette to be 184 ATP per molecule (metabolic costs for precursors were disregarded in this estimate); see *Keller and Surette, 2006* for details. In contrast, basically no costs (0–1 ATP) incur for the different signaling molecule Autoinducer-2 (AI-2) that is considered as a metabolic by-product. As to what extent the production of oligopeptides for signaling reduces an individual's growth rate has, to our knowledge, not been studied quantitatively.

For quorum-sensing systems that involve $N$-acyl homoserine lactones (AHLs) as signaling molecules, however, a reduced fitness of producers has been reported for microbial growth in batch culture (*Ruparell et al., 2016*; *Diggle et al., 2007*; *He et al., 2003*). Even though metabolic costs for the synthesis of $C_4$-HSL (one of the simplest AHL signaling molecules that is synthesized via the *rhl* operon) were conservatively estimated with only 8 ATP per molecule (*Keller and Surette, 2006*), a growth impairment was experimentally reported only recently for a $C_4$-HSL-producing strain (*Ruparell et al., 2016*). Furthermore, a strain producing a long-chain AHL ($OC_{12}$-HSL, synthesized via the *las* operon) showed a reduced fitness in both mono and mixed culture compared with a non-producing strain. The reduced fitness of AHL-producers was attributed to (i) metabolic costs of autoinducer production, in particular also to metabolic costs of precursors that were disregarded in the estimates by *Keller and Surette, 2006*, and (ii) accumulation of toxic side products accompanying the synthesis of autoinducers (*Ruparell et al., 2016*). As another example, the strain *Sinorhizobium fredii* NGR234 synthesizes AHLs via both the *ngr* and the *tra* operon (*Schmeisser et al., 2009*), and it was shown that gene expression related to autoinducer production reduces the strain's growth rate in mono culture (*He et al., 2003*). On the other hand, a heterogeneous expression of the corresponding autoinducer synthase genes was observed during growth of NGR234 only recently (*Grote et al., 2014*). As to what extent the production of AHLs reduces fitness of NGR234 in mixed culture and, thus, whether the phenotypic heterogeneity observed in *Grote et al., 2014* could be explained through the ecological feedback proposed by our quorum-sensing model, remains to be explored experimentally.

In the quorum-sensing model, even small growth rate differences between producer and non-producer, which are quantified by the ratio (growth rate of producer) / (growth rate of non-producer) $= 1 - s$, may give rise to a bimodal production of autoinducers in the population. Furthermore, it would be interesting to track the expression level of autoinducer synthase genes of a microbial strain

during growth for which growth differences between the producing and the non-producing pheno-type are known such as in the study of *Ruparell et al., 2016*. We emphasize that it would be desir-able to report the full distribution of expression levels in the population in order to detect whether a population splits into several subpopulations; note that variance or percentiles are not suitable measures to characterize and compare the bimodality of distributions. A bimodal expression of auto-inducer synthase genes in the population together with a tightly controlled average expression level could be a signature of the feedback between ecological and population dynamics underlying the observation of phenotypic heterogeneity as suggested by our results.

## A question of spatio-temporal scales: How stable and how dispersed are autoinducers in the environment?

Autoinducers are secreted into the environment where they get dispersed and are degraded. For simplicity and to facilitate our mathematical analysis, we assumed in the quorum-sensing model that individuals respond to the current average production level of autoinducers in the whole population. Temporal availability and spatial dispersal of autoinducers determine whether this assumption is valid or not. On the one hand, temporal availability of autoinducers in the environment for signaling depends on many factors. For example, pH and temperature influence the stability of autoinducers (*Yates et al., 2002*; *Byers et al., 2002*; *Decho et al., 2009*; *Grandclément et al., 2016*; *Hmelo, 2017*). Biochemical mechanisms that inhibit or disrupt the functioning of signaling molecules (commonly referred to as 'quorum quenching') further determine the time scales at which autoin-ducers are degraded in the environment (*LaSarre and Federle, 2013*; *Grandclément et al., 2016*; *Hmelo, 2017*). On the other hand, spatial dispersal of autoinducers in the population depends, for example, upon cellular mechanisms that import and export autoinducers into the cell from the envi-ronment and vice versa, and upon the spatial structure of the microbial population (*Platt and Fuqua, 2010*; *Hense and Schuster, 2015*). The degree of dispersal determines whether autoinducers remain spatially privatized to a single cell, diffuse to neighboring cells, or are spread evenly between all cells of the population. Consequently, the spatio-temporal organization of the microbial popula-tion determines as to what extent microbes sense rather the current average production level or a time-integrated production of autoinducers, and to what extent they sense rather the global or a local average production level. Our quorum-sensing model assumes that autoinducers are uniformly degraded in a well-mixed environment. These assumptions do not hold true for a spatially structured microbial biofilm, but should be fulfilled during the stationary phase of microbial growth in a well-mixed batch culture (*Yates et al., 2002*; *Byers et al., 2002*).

## How is production of autoinducers up-regulated at the single-cell level?

### Monostable or bistable up-regulation of autoinducer synthesis at the single-cell level

Our theoretical results also relate to the question of how cells regulate the production of autoin-ducers upon sensing the level of autoinducers in the environment. In this work, we showed that posi-tive feedback loops and, thus, up-regulation of cellular autoinducer production may give rise to phenotypic heterogeneity. Positive feedback loops are mathematically introduced in our model as a stable fixed point at the producing phenotype of the response function (up-regulation to the stable 'on' state at $p = 1$; see *Figure 1B*). Such a positive feedback is not present in all autoinducer syn-thase systems, but was reported for the strains *L. monocytogenes* and *S. fredii* NGR234 (*Waters and Bassler, 2005*; *Garmyn et al., 2009*; *He et al., 2003*; *González and Marketon, 2003*) that showed a heterogeneous synthesis of autoinducers at the population level (*Garmyn et al., 2011*; *Grote et al., 2014*). From an experimental point of view it is often not known, however, whether autoinducer synthesis is up-regulated for all autoinducer levels or only above a threshold level. Up-regulation at all production levels in the population corresponds to a monostable response function with an unstable fixed point at the 'off' state at $p = 0$, whereas up-regulation only above a threshold level corresponds to a bistable response function with a stable fixed point at the 'off' state at $p = 0$ and an additional unstable fixed point at the threshold value (see *Figure 1B*). Most models of quorum-sensing microbial populations explicitly or implicitly assume a bistable gene regulation for positive feedback loops without experimental verification; see (*Hense and Schuster, 2015*) for further discussion. Why might it be relevant to distinguish between bistable (for example, a Hill

function with Hill coefficient > 1) and monostable (for example, a Hill function with Hill coefficient ≤ 1) regulation of autoinducer synthesis – apart from the insight on how regulation proceeds at the molecular level? As the results of our quorum-sensing model show, the qualitative form of the regulation could discriminate between different mechanisms that control phenotypic heterogeneity of the autoinducer production at the population level as we describe in the following.

## Heterogeneity through stochastic gene expression only for bistable gene regulation

In recent years, a deeper mechanistic understanding of phenotypic heterogeneity has been achieved by exploring how the presence of different phenotypes in a population of genetically identical cells depends upon molecular mechanisms and stochasticity at the cellular level (*Ackermann, 2015*). For example, a bistable gene regulation function enables cells to switch between an 'on' and an 'off' state with respect to the expression of a certain gene or operon. Depending on environmental cues, cells are either in the stable 'on' or in the stable 'off' state. A noisy expression at intermediate concentrations of an environmental cue may then cause some cells to be in the 'on' state while others are still in the 'off' state. Thus, stochastic gene expression explains the coexistence of different phenotypic states in one population in many experimental situations (*Novick and Weiner, 1957*; *Ozbudak et al., 2004*; *Kaern et al., 2005*; *Dubnau and Losick, 2006*; *Smits et al., 2006*; *Raj and van Oudenaarden, 2008*; *Eldar and Elowitz, 2010*). In the context of quorum sensing, the level of autoinducers in the population is the environmental cue that triggers the stochastic switch between 'on' and 'off' state explaining heterogeneous autoinducer production when the response function is bistable (*Fujimoto and Sawai, 2013*; *Pérez-Velázquez et al., 2016*; *Goryachev et al., 2005*; *Dockery and Keener, 2001*). In other words, bistable regulation together with stochastic gene expression can explain a bimodal autoinducer synthesis in the population. If, however, regulation of autoinducer synthesis is monostable, an explanation of phenotypic heterogeneity in the autoinducer production in terms of stochastic gene expression appears questionable to us.

## Heterogeneity through an ecological feedback for monostable and for bistable gene regulation

The analysis of our quorum-sensing model suggests that an alternative mechanism could explain a heterogeneous production of autoinducers in quorum-sensing microbial populations. Our results show that phenotypic heterogeneity may also arise dynamically as a collective phenomenon for monostable regulation of autoinducer production when quorum sensing creates an ecological feedback by coupling ecological with population dynamics. Cells need to up-regulate their expression with respect to the sensed production level in the population. A threshold-like, bistable response function does not need to be assumed in the quorum-sensing model, but would work as well, to establish a bimodal production of autoinducers in the population.

Therefore, if phenotypic heterogeneity of autoinducer synthesis is observed in a microbial population and if cellular growth rate is correlated with the cell's production degree of autoinducers, then it would be worth testing experimentally whether regulation of autoinducer synthesis is monostable or bistable. Monostable regulation would be an indicator that heterogeneity on the population level is not caused by stochastic gene expression, but actually is caused by a different mechanism such as the ecological feedback proposed here.

## On which timescales do microbes respond to autoinducers in the environment?

Furthermore, in our implementation of the quorum-sensing model, individuals respond to the environment with response probability $\lambda$ upon reproduction. The rule that offspring individuals can only respond at reproduction events represents a coarse-grained view in time to facilitate the mathematical analysis and to identify the ecological feedback. The response probability can actually be interpreted as the rate with which individuals respond to autoinducers in the environment. This cellular response rate is then effectively measured in units of the cell's reproduction rate ($\phi_i$) in the quorum-sensing model. Phenotypic heterogeneity of autoinducer production arises in the quorum-sensing model if the time scale at which cells respond to autoinducers in the environment is of similar order as or larger than the time scale at which growth rate differences affect the population dynamics. This

can be inferred from the prefactors of the sense-and-response term and the replicator term in the autoinducer equation *(1)*: *Effective changes* of the distribution of autoinducer production in the population occur (i) through cellular response to autoinducers in the environment at rate ~$2\lambda$ and (ii) through growth rate differences at rate ~$s$. Both contributions need to balance each other such that a bimodal production in the population is established (quantified in our model by the ratio $\beta = 2\lambda/s$; see also *Box 1* for an illustration). This balance is robust against several kinds of perturbations and noise as discussed above; see *Appendix 1—figures 1* and *2*. To understand how bacteria respond to changes of autoinducer levels in the environment and to quantify response rates, experiments at the single-cell level seem most promising to us at present.

## Single-cell experiments

Some of the questions raised above may be addressed most effectively with single-cell experiments. For example, it would be desirable to simultaneously monitor, at the single-cell level, the correlations between autoinducer levels in the environment, the expression of autoinducer synthase genes, and the transcriptional regulators that mediate response to quorum sensing. Upon adjusting the level of autoinducers in a controlled manner, for example in a microfluidic device, one could characterize how cells respond to autoinducers in the environment. This way, it might be possible to answer questions of (i) how the cellular production of autoinducers is regulated (monostable or bistable regulation, or a different form of regulation), (ii) whether response times to environmental changes are stochastic and whether response rates can be identified, (iii) as to what extent cellular response in the production of autoinducers depends on both the level of autoinducers in the environment and on the cell's present production degree, and (iv) how production of autoinducers is correlated with single-cell growth rate. In the context of the quorum-sensing model, the results of such single-cell experiments would help to identify the form of the fitness function $\phi$ and the response function $R$, to quantify the selection strength $s$ and response probability $\lambda$, and to refine the model set-up.

Different mechanisms at the cellular (microscopic) level may yield the same behavior at the population (macroscopic) level. Therefore, observations at the population level might not discriminate between different mechanisms at the cellular level. Is phenotypic heterogeneity in the production of autoinducers an example of such a case? In this work, we discussed that phenotypic heterogeneity in the autoinducer production could be the result of stochastic gene expression in bistable gene regulation or, as suggested by our model, the result of the feedback between ecological and population dynamics. We believe that the above-mentioned single-cell experiments could elucidate the mechanisms that allow for phenotypic heterogeneity in quorum-sensing microbial populations, and help to understand how population dynamics and ecological dynamics influence each other.

## What is the function of phenotypic heterogeneity in autoinducer production?

The purpose of the quorum-sensing model presented here is to explain how phenotypic heterogeneity in the autoinducer production arises and how it is controlled in quorum-sensing microbial populations. With the current model set-up, however, we did not address its function. Why might this phenotypic heterogeneity in the autoinducer production be beneficial for a microbial species on long times? From an experimental point of view, the evolutionary contexts and ecological scenarios under which this phenotypic heterogeneity may have arisen are still under investigation (*Garmyn et al., 2011*; *Grote et al., 2014*, *Grote et al., 2015*). From a modeling perspective, one could extend, for example, our chosen fitness function with a term that explicitly accounts for the benefit of signaling either at the cellular or population level, and study suitable evolutionary contexts and possible ecological scenarios (*Pollak et al., 2016*; *Dandekar et al., 2012*; *Czárán and Hoekstra, 2009*; *Carnes et al., 2010*; *Hense and Schuster, 2015*). Such theoretical models together with further experiments might help to clarify whether heterogeneous production of autoinducers can be regarded as a bet-hedging strategy of the population or rather serves the division of labor in the population (*Ackermann, 2015*).

## Conclusion

Overall, our analyses suggest that feedbacks between ecological and population dynamics through signaling might generate phenotypic heterogeneity in the production of signaling molecules itself, providing an alternative mechanism to stochastic gene expression in bistable gene-regulatory circuits. Spatio-temporal scales are important for the identified ecological feedback to be of relevance for microbial population dynamics: growth rate differences between producers and non-producers need to balance the rate at which cells respond to the environment, degradation of signaling molecules should be faster than time scales at which growth rate differences affect the population composition significantly, and signaling molecules should get dispersed in the whole population faster than they are degraded. In total, if microbes sense and respond to their self-shaped environment under these conditions, the population may not only respond as a homogeneous collective as is typically associated with quorum sensing, but may also become a robustly controlled heterogeneous collective. Further experimental and theoretical studies are needed to clarify the relevance of the different mechanisms that might control phenotypic heterogeneity, in particular for quorum-sensing microbial populations.

# Materials and methods

### Derivation of the autoinducer equation *(1)*

The microscopic dynamics are captured by a memoryless stochastic birth-death process (a continuous-time Markov process) as sketched in *Figure 1*. The state of the population **p** is updated by non-genetic inheritance and sense-and-response through quorum sensing such that at most two individuals $i$ and $j \neq i$ change their production degree at one time. The temporal evolution of the corresponding joint $N$-particle probability distribution $P(\mathbf{p}, t)$ is governed by a master equation for the stochastic many-particle process (*Gardiner, 2009*; *Van Kampen, 2007*; *Weber and Frey, 2017*), whose explicit form is derived from *Figure 1* and given in *Appendix 2*. This master equation tracks the correlated microscopic dynamics of the production degrees of all $N$ individuals. To make analytical progress, we focused on the reduced one-particle probability distribution $\rho^{(1)}(p, t) = 1/N \langle \sum_i \delta(p - p_i) \rangle_P$ in the spirit of a kinetic theory (*Kadar, 2007*) starting from the microscopic stochastic dynamics. $\rho^{(1)}$ denotes the probability distribution of finding *any* individual at a specified production degree $p$ at time $t$; the numerically obtained histogram of $\rho^{(1)}$ was plotted in *Figure 2*. The temporal evolution of $\rho^{(1)}$ is derived from the master equation, and couples to the reduced two-particle probability distribution and to the full probability distribution $P$ through quorum sensing. By assuming that correlations are negligible, one may approximate $\rho^{(1)}$ by the mean-field distribution $\rho$, which we refer to as the production distribution. The mean-field equation *(1)* for $\rho$ is derived in *Appendix 2* and referred to as the autoinducer equation. Note that *Equation (1)* conserves normalization of $\rho$, that is, $\int_0^1 \mathrm{d}p \, \partial_t \rho(p, t) = 0$.

We also proved that $\rho^{(1)}$ converges in probability to $\rho$ as $N \to \infty$ for any finite time if initial correlations are not too strong. In other words, the autoinducer equation *(1)* captures exactly the collective dynamics of the stochastic many-particle process for large $N$. To show this convergence, we introduced the bounded Lipschitz distance $d$ between $\rho$ and $\rho^{(1)}$, applied Grönwall's inequality to the temporal evolution of $d$, and used the law of large numbers; see (*Frey et al., 2017*) for details. Similar distance measures and estimates have been used, for example, to prove that the Vlasov equation governs the macroscopic dynamics of the above-mentioned classical XY spin model with infinite range interactions (*Braun and Hepp, 1977*; *Dobrushin, 1979*; *Spohn, 1991*; *Yamaguchi et al., 2004*).

### Analysis of homogeneous stationary distributions of the autoinducer equation *(1)*

Without quorum sensing ($\lambda = 0$), one finds the analytical solution for $\rho$ by applying the method of characteristics to *Equation (1)* in the space of moment and cumulant generating functions as: $\rho(p, t) = \rho_0(p)e^{-stp} / \int_0^1 \mathrm{d}p \, e^{-stp}\rho_0(p)$; see *Appendix 3* for details. Thus, the initially lowest production degree in the population, $p_{\text{low}}$, constitutes the homogeneous stationary distribution

$\rho_\infty(p) = \delta(p - p_{\text{low}})$, which is attractive for generic initial conditions. Only $\delta$-peaks at production degrees greater than $p_{\text{low}}$ are stationary as well, but they are neither attractive nor stable. The temporal approach to the homogeneous stationary distribution is algebraically slow for continuous initial distributions $\rho_0$, and exponentially fast if $p_{\text{low}}$ is separated from all greater degrees by a gap in production space; see *Appendix 3* and *Figure 2D*.

With quorum sensing ($\lambda > 0$), fixed points of the response function $p^* = R(p^*)$ yield homogeneous stationary distributions of the autoinducer equation *(1)* as $\rho_\infty(p) = \delta(p - p^*)$. In particular, stable fixed points of the response function ($R'(p^*) < 1$) constitute homogeneous stationary distributions that are stable up to linear order in perturbations around stationarity. For $\lambda > s/2$, these distributions are also attractors of the mean-field dynamics *(1)* for all initial distributions; see *Appendix 3*. The temporal approach towards homogeneous stationary distributions with quorum sensing is generically exponentially fast (*Figure 2E*). This exponentially fast approach is illustrated for the special case of a linear response function and $\lambda = 1/2$, for which one finds the analytical solution as: $\rho(p, t) = y(t)\rho_0(p) + (1 - y(t))\delta(p - \overline{p}_0)$ with $y(t) = e^{-\overline{\phi}_0 t}$. However, time scales at which stationarity is approached may diverge at bifurcations of the response function. Such can be seen, for example, if one chooses a supercritical pitchfork bifurcation of a polynomial response function and $\lambda = 1/2$; see *Appendix 1—figure 3* and *Appendix 3*.

## Phase transitions from heterogeneity to homogeneity in the autoinducer equation *(1)*

Here we discuss how the long-time behavior of the quorum-sensing model changes from heterogeneous to homogeneous populations as the response probability $\lambda$ vanishes or reaches the upper threshold $\lambda_{\text{up}}$ while the selection strength $s$ is kept fixed. For small response probabilities, $0 < \lambda < \lambda_{\text{up}}$, the heterogeneous stationary distributions of the autoinducer equation *(1)* explain the long-lived, heterogeneous states of the stochastic quorum-sensing process. The coexisting $\delta$-peaks at the low-producing and high-producing degree in the heterogeneous stationary distribution are separated by a gap in production space, which gives rise to the non-vanishing variance $\text{Var}(p)_\infty$ in the phase of heterogeneity (*Figure 3C*). As $\lambda \to \lambda_{\text{up}}$, the gap closes, $p_{\text{high}} \to R(p_{\text{high}})$, and $y \to 0$, such that a homogeneous stationary distribution with $\text{Var}(p)_\infty = 0$ is recovered in a continuous transition. This non-equilibrium phase transition from heterogeneity to homogeneity proceeds without any critical behavior. As $\lambda \to 0$, and under the assumption that 0 is an unstable fixed point of the response function ($R(0) = 0$ and $1 < R'(0)$; we further assume $R'(0) < \infty$), the gap between the low-producing and the high-producing peak closes as well because $p_{\text{high}} \to 0$. However, $y$ does not approach 1, but the value $1 - 1/R'(0) < 1$. The probability weight at the low-producing mode jumps by the value $1/R'(0)$ and the homogeneous stationary distribution with $\text{Var}(p)_\infty = 0$ is recovered in a discontinuous transition. Therefore, a discontinuous phase transition in the space of stationary probability distributions governs the long-time dynamics of the autoinducer equation *(1)* from heterogeneity to homogeneity as the response probability $\lambda$ vanishes (for fixed selection strength $s$).

## Acknowledgements

We thank the QBio 2014 summer course 'Microbial Strategies for Survival and Evolution' at the Kavli Institute for Theoretical Physics at the University of California, Santa Barbara, from which this work originated. We acknowledge fruitful discussions on phenotypic heterogeneity and quorum sensing with Paul Rainey, Kirsten Jung, Kai Papenfort, Wolfgang Streit, Jessica Grote, Vera Bettenworth, Friedrich Simmel, and Madeleine Opitz. We also thank Mauro Mobilia, Meike Wittmann, Florian Gartner, Markus F. Weber, Karl Wienand, Jonathan Liu, and Alexander Dobrinevski for discussions on the quorum-sensing model. MB appreciates funding by a Qualcomm European Research Studentship. This research was supported by the German Excellence Initiative via the program 'Nanosystems Initiative Munich' (NIM) and by the Deutsche Forschungsgemeinschaft within the framework SPP1617 (through grant FR 850/11-1, 2) on phenotypic heterogeneity and sociobiology of bacterial populations. MB, JK, ML, PP, and EF designed research, performed research, and wrote the paper. The authors declare no conflict of interest.

# Additional information

## Funding

| Funder | Grant reference number | Author |
| --- | --- | --- |
| Deutsche Forschungsgemeinschaft | SPP1617 through grant FR850/11-1, 2 | Matthias Bauer<br>Johannes Knebel<br>Matthias Lechner<br>Erwin Frey |
| German Excellence Initiative | Nanosystems Initiative Munich | Matthias Bauer<br>Johannes Knebel<br>Matthias Lechner<br>Erwin Frey |
| Qualcomm European Research Studentship | n/a | Matthias Bauer |

The funders had no role in study design, data collection and interpretation, or the decision to submit the work for publication.

## Author contributions

MB, Conceptualization, Data curation, Software, Formal analysis, Validation, Investigation, Visualization, Methodology, Writing—original draft, Project administration, Writing—review and editing; JK, Conceptualization, Data curation, Formal analysis, Validation, Investigation, Visualization, Methodology, Writing—original draft, Writing—review and editing, project administration; ML, Data curation, Software, Visualization, Methodology, Writing—review and editing; PP, Formal analysis, Methodology, Writing—review and editing; EF, Conceptualization, Resources, Data curation, Software, Formal analysis, Supervision, Funding acquisition, Validation, Investigation, Visualization, Methodology, Writing—original draft, Project administration, Writing—review and editing

## Author ORCIDs

Matthias Bauer, http://orcid.org/0000-0001-7040-2054

Johannes Knebel, http://orcid.org/0000-0002-0660-9496

Matthias Lechner, http://orcid.org/0000-0002-0452-7091

Erwin Frey, http://orcid.org/0000-0001-8792-3358

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

# Appendix 1

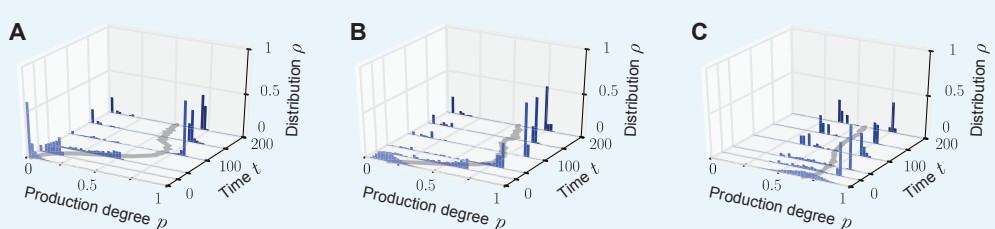

**Appendix 1—figure 1.** Phenotypic heterogeneity in the quorum-sensing model arises for diverse initial distributions. Bimodal quasi-stationary states arise for a broad class of initial distributions if the value of the response probability $\lambda$ is small and an individual's production degree is upregulated by the sense-and-response mechanism through quorum sensing ($R(p) > p$ for some $p \in [0, 1]$). Depicted is the temporal evolution of the histograms of production degrees (normalized values) as in *Figure 2* of the main text. The monostable response function $R(p) = p + 0.2 \cdot \sin(\pi p)$ was chosen (see *Figure 1B*). (**A, B**) $\lambda = 0.05$. Initially, the population consists of mainly non-producers (in (**A**) initial distribution $p_i \sim \mathrm{Beta}(0.5, 20)$ i.i.d. and in (**B**) initial distribution $p_i \sim \mathrm{Beta}(4, 20)$ i.i.d.). Due to the balance of fitness differences and sense-and-response through quorum sensing, the population splits into a heterogeneous population with producers and non-producers coexisting for long times. (**C**) $\lambda = 0.02$. If the initial distribution of production degrees is centered around high production degrees (initial distribution $p_i \sim \mathrm{Beta}(10, 5)$ i.i.d.), the population may still evolve in time into a heterogeneous quasi-stationary state. However, the peak at the low-producing degree is typically located away from 0, that is, $p_{\mathrm{low}} > 0$. These exemplary numerical results (**A–C**) are confirmed by the results of our mean-field theory: heterogeneous stationary distributions are the attractor of the mean-field dynamics (autoinducer equation *(1)* in the main text) for a broad range of initial distributions if conditions (i) $R(\bar{p}_\infty) = p_{\mathrm{high}} > \bar{p}_\infty$ and (ii) $\lambda < \lambda_{\mathrm{up}} = s/2$ are fulfilled (see main text). Note that 'i.i.d.' abbreviates 'independent and identically distributed'. *Parameters:* selection strength $s = 0.2$ and population size $N = 10^4$.

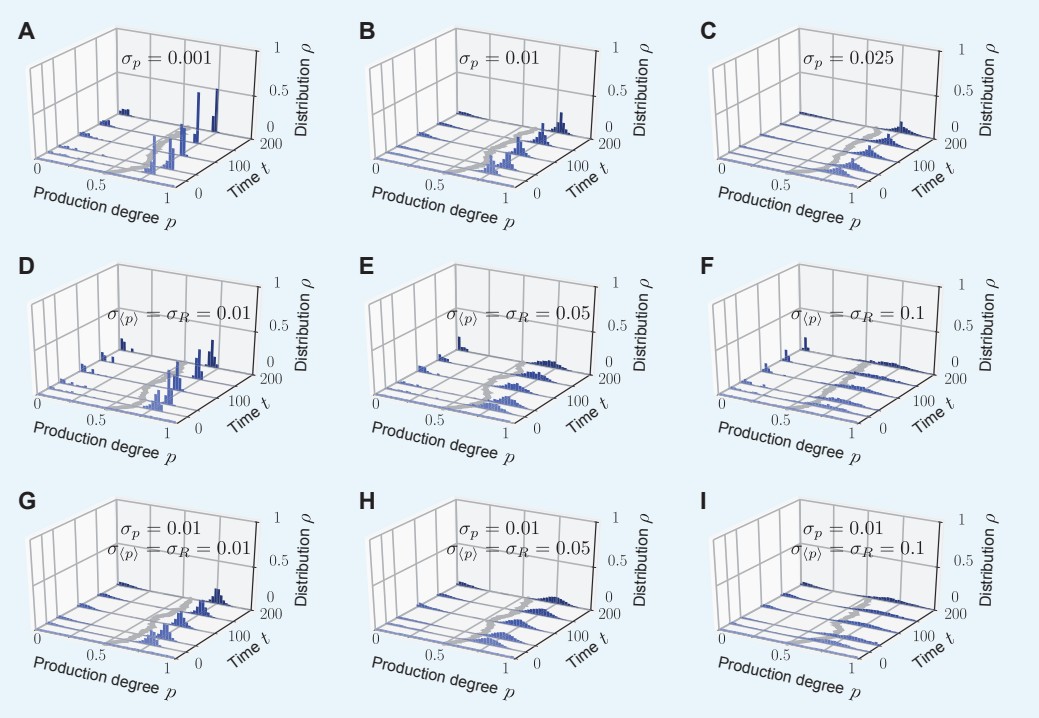

**Appendix 1—figure 2.** Phenotypic heterogeneity in the quorum-sensing model is robust against noisy inheritance, noisy perception, and noisy response. Upon including either noisy inheritance of the production degree (**A–C**), or noisy perception of the average production level and noisy response to it (**D–F**), or both percx=" ">(**G-I**) into the model set-up, bimodal quasi-stationary states still arise in the relevant parameter regimes (see *Figure 2C*). Depicted are representative single realizations of the modified stochastic process (histogram over normalized values of production degrees to make the comparison with *Figure 2* possible). (**A–C**) Noisy inheritance is implemented at reproduction events. Production degree $p_i$ is passed on to an offspring as $p_i \mapsto p_i + \eta_p$ with noise $\eta_p \sim \mathcal{N}(0, \sigma_p)$ sampled from a Normal distribution (and are cut off such that $p_i + \eta_p \in [0,1]$), emulating noisy inheritance of the phenotype. $\sigma_p \geq 0$ characterizes the strength of the noise ($\sigma_p = 0$ recovers noiseless inheritance). As $\sigma_p$ increases, bimodal quasi-stationary states still arise, but the two peaks become broader than in the noiseless case. (**D–F**) Noise in the sensing apparatus is implemented as noisy perception of the average production level $\langle p \rangle \mapsto \langle p \rangle + \eta_{\langle p \rangle}$ with Gaussian noise $\eta_{\langle p \rangle} \sim \mathcal{N}(0, \sigma_{\langle p \rangle})$, and noise in the response is implemented at the level of the response function as $R(\langle p \rangle) \mapsto R(\langle p \rangle) + \eta_R$ with Gaussian noise $\eta_R \sim \mathcal{N}(0, \sigma_R)$. Therefore, the production degree of an individual is updated through sense-and-response to the environment as $p_i = R(\langle p \rangle) \mapsto R(\langle p \rangle + \eta_{\langle p \rangle}) + \eta_R$ in the quorum-sensing model. Again, as the strength of both sense and response noise increase, bimodal quasi-stationary states still arise, but the two peaks become broadened compared with the noiseless case. We emphasize that $\sigma_{\langle p \rangle} = \sigma_R = 0.1$ corresponds to very strong noise on the interval $[0,1]$. (**G–I**) Combined effect of noisy inheritance and noisy sense-and-response. Representative trajectories demonstrate that bimodal quasi-stationary states also arise in the presence of noise at all update steps. Thus, phenotypic heterogeneity in the quorum-sensing model is qualitatively robust against noise at all steps. *Initial distribution:* $p_i \sim \mathrm{Uniform}(0,1)$, independent and identically distributed; *Parameters:* selection strength $s = 0.2$, response probability $\lambda = 0.05$, response function $R(\langle p \rangle) = \langle p \rangle + 0.2 \cdot \sin(\pi \langle p \rangle)$, and population size $N = 10^4$.

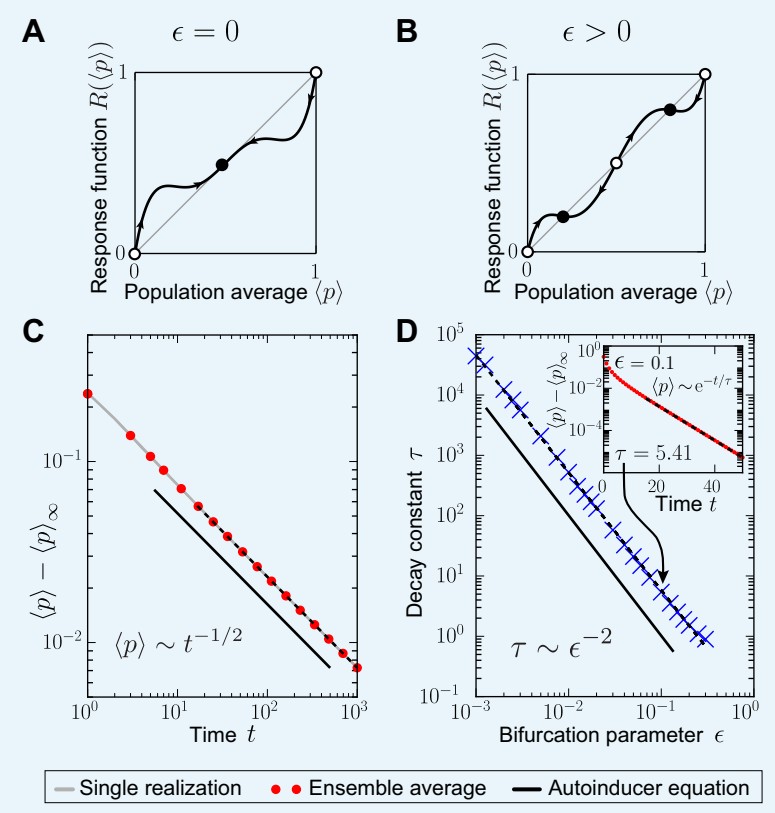

**Appendix 1—figure 3.** Time scales at which stationarity is approached may diverge. The response probability was set to $\lambda = 1/2$, and the nonlinear response function $R(\langle p \rangle) = \langle p \rangle + 40 \cdot \langle p \rangle (\langle p \rangle - (0.5 - \epsilon))(\langle p \rangle - 0.5)(\langle p \rangle - (0.5 + \epsilon))(\langle p \rangle - 1)$ with bifurcation parameter $\epsilon$ was chosen, see *Equation (48)*; $\epsilon$ controls a supercritical pitchfork bifurcation of the response function at the fixed point $p_{cr} = 0.5$ ($R(p_{cr}) = p_{cr}$): For $\epsilon > 0$, the fixed point at $p_{cr}$ is unstable and non-degenerate (sketch in (**B**)), and becomes stable and threefold degenerate ($z = 3$) as $\epsilon = 0$ (sketch in (**A**)). (**D**) Away from the bifurcation of the response function ($\epsilon > 0$), the approach of an absorbing state in the stochastic many-particle system is exponentially fast (see inset of (**D**) for an exemplary measurement of $\langle p \rangle (t) - \langle p \rangle_\infty$ for $\epsilon = 0.1$, dashed line denotes fit to exponential decay). The exponentially fast approach of stationarity is confirmed by mean-field theory ($\bar{p}_t - \bar{p}_\infty \sim e^{-t/\tau}$), see main text and *Equation (51)*. Mean-field theory also predicts that the time scale of this exponentially fast relaxation diverges as $\tau \sim \epsilon^{-2}$ as the bifurcation is approached ($\epsilon \to 0$), indicated by the black line in (**D**). This prediction agrees with the numerical simulations of the stochastic quorum-sensing model, see (**D**) (blue crosses denote values of the decay constants obtained from the exponential fits and black dashed line indicates fit to $\tau \sim 1/\epsilon^\gamma$ with $\gamma = 1.95$). The divergence of time scales reflects critical slowing down as $\epsilon \to 0$. (**C**) At the bifurcation of the response function ($\epsilon = 0$), the approach of an absorbing state is algebraically slow, $\bar{p}_t - \bar{p}_\infty \sim t^{-1/\nu}$ with critical exponent $\nu = z - 1 = 2$ obtained from mean-field theory (black line), see *Equation (53)*. This prediction agrees with our numerical simulations of the stochastic quorum-sensing model (black dashed line in (**C**) indicates fit to $\langle p \rangle (t) - \langle p \rangle_\infty \sim t^\alpha$ with $\alpha = -0.50$). *Initial distribution:* unimodal $p_i \sim \text{Beta}(1, 10)$, independent and identically distributed; *Parameters:* Ensemble size $M = 100$, selection strength $s = 0.1$, population size $N = 10^4$.

## Appendix 2

# From a microscopic description to a macroscopic description of the quorum-sensing model

### Description of the microscopic dynamics: Master equation of the stochastic many-particle process

To describe the temporal evolution of the population, we introduced the joint $N$-particle probability distribution $P(\mathbf{p}, t)$. The value $P(\mathbf{p}, t)\mathrm{d}p_1 \ldots \mathrm{d}p_N$ denotes the joint probability of finding the first individual with a production degree in the interval $[p_1, p_1 + \mathrm{d}p_1]$, the second individual with a production degree in the interval $[p_2, p_2 + \mathrm{d}p_2]$, and so on at time $t$. The stochastic dynamics are captured by a coupled birth-death process (continuous-time Markov process) as described in the main text and in *Figure 1* of the main text. An individual $i$ reproduces randomly after a time that is exponentially distributed with rate $\phi_i$, which we refer to as the individual's fitness in the main text. One update step involves reproduction, sense-and-response through quorum sensing, and non-genetic inheritance such that at most two individuals $i$ and $j \neq i$ change their production degree at one time. We denote the state of the population before the update step as $\widetilde{\mathbf{p}}^{i,j} = (p_1, \ldots, p_{i-1}, \widetilde{p}_i, p_{i+1} \ldots, \widetilde{p}_j, \ldots, p_N)$; the production degrees of individual $i$ and $j$, which might change during the update step, are labeled with a tilde. For the sake of readability, we do not distinguish notationally between a random variable and the value that this random variable attains; both are labeled with the same symbol. The master equation for the joint $N$-particle probability distribution $P$ for the individuals' production degrees $\mathbf{p} = (p_1, \ldots, p_N)$ at time $t$ can be written as (*Gardiner, 2009*; *Van Kampen, 2007*; *Weber and Frey, 2017*):

$$
\begin{aligned}
\partial_t P(\mathbf{p}, t) \; &= \sum_{i=1}^{N} \sum_{j \neq i}^{N} \int_{[0,1]^2} \mathrm{d}\widetilde{p}_i \mathrm{d}\widetilde{p}_j \, P(\widetilde{\mathbf{p}}^{i,j}, t) \phi_i(\widetilde{\mathbf{p}}^{i,j}) \psi_j(\widetilde{\mathbf{p}}^{i,j}) A_i(\widetilde{\mathbf{p}}^{i,j}; i) A_j(\widetilde{\mathbf{p}}^{i,j}; i) \\
&\quad - P(\mathbf{p}, t) \sum_{i=1}^{N} \sum_{j \neq i}^{N} \phi_i(\mathbf{p}) \psi_j(\mathbf{p}) \, , \\
&= \mathrm{gain} - \mathrm{loss} \, ,
\end{aligned}
\tag{3}
$$

with reproduction rate of individual $i$ (fitness) given by (and selection strength $0 \leq s < 1$):

$$
\phi_i(\mathbf{p}) = \phi(p_i) = 1 - sp_i \, ,
\tag{4}
$$

and death rate of individual $j$ given by (random death):

$$
\psi_j(\mathbf{p}) = \frac{1}{N-1} \, .
\tag{5}
$$

The transition probabilities $A_i$ and $A_j$ account for the ensuing changes of at most two production degrees in the population due to non-genetic inheritance and sense-and-response (see detailed description below *Equation (6)* and *Equation (7)*). The initial condition to the master equation *(3)* is given as $P(\mathbf{p}, t = 0) = p_0(\mathbf{p})$.

The master equation *(3)* involves two contributions: *gain terms* yielding an increase and *loss terms* yielding a decrease of the probability weight in state $\mathbf{p}$ at time $t$. *Loss terms* occur when the population is in state $\mathbf{p}$ and an individual reproduces. The probability of finding the population in this state is given by $P(\mathbf{p}, t)$. Individual $i$ is selected for reproduction at rate $\phi_i(\mathbf{p})$ and splits into two offspring individuals, and a different individual $j \neq i$ is removed with probability $1/(N-1)$ at the same time (random death). *Gain terms* involve all events that

take the population from an arbitrary state $\widetilde{\mathbf{p}}^{i,j}$ to state $\mathbf{p}$, and involve again reproduction for individual $i$ and neutral death for individual $j$. The transition probabilities $A_i$ and $A_j$ account for these changes due to non-genetic inheritance and sense-and-response through quorum sensing, and are given as:

$$A_i(\widetilde{\mathbf{p}}^{i,j}; i) = \lambda \cdot \delta\left(p_i - R(\langle \widetilde{p}^{i,j} \rangle)\right) + (1-\lambda) \cdot \delta(p_i - \widetilde{p}_i), \tag{6}$$

$$A_j(\widetilde{\mathbf{p}}^{i,j}; i) = \lambda \cdot \delta\left(p_j - R(\langle \widetilde{p}^{i,j} \rangle)\right) + (1-\lambda) \cdot \delta(p_j - \widetilde{p}_i). \tag{7}$$

We abbreviate $\langle \widetilde{p}^{i,j} \rangle = 1/N \sum_k (\widetilde{\mathbf{p}}^{i,j})_k$ as the average production degree before the update step. Both transition probabilities $A_i$ and $A_j$ quantify the probability of attaining the production degrees $p_i$ and $p_j$, respectively, for the two offspring individuals of ancestor $i$. The first summand in both $A_i$ and $A_j$ captures the response to the perceived average production ($p_{i/j}$ attains the value $R(\langle \widetilde{p}^{i,j} \rangle)$) with probability $\lambda$ as the updated production degree, and the second summand accounts for the non-genetic inheritance of the production degree from the ancestor $i$ ($p_{i/j}$ attains the value $\widetilde{p}_i$) with probability $1-\lambda$. Note that for the transition probability $A_j$, also $p_j$ attains the value $\widetilde{p}_i$ due to our convention that individual $i$ is labeled as the reproducing individual and individual $j$ is chosen for the death event, see **Figure 1** of the main text.

In our prescription of the master equation **(3)**, the introduced gain and loss terms also involve terms that actually do not change the state of the population. Such is the case, for example, when the two individuals $i$ and $j$ have the same production degree ($\widetilde{p}_i = \widetilde{p}_j$) and both offspring individuals retain the production degree from their ancestor $i$ ($p_i = p_j = \widetilde{p}_i$, that is, both offspring individuals do not update their production through sense-and-response). Such events do not change the state of the population ($\widetilde{\mathbf{p}}^{i,j} = \mathbf{p}$), but are included in the master equation **(3)**. However, these terms always occur both in the gain and loss terms. Therefore, they cancel each other and the master equation can be written in form of **Equation (3)**.

The master equation **(3)** conserves normalization of $P$ because $\partial_t \int_{[0,1]^N} \mathrm{d}\mathbf{p}\, P(\mathbf{p}, t) = 0$; see analysis below.

## Coarse-grained description: Reduced one-particle probability distribution

The reduced one-particle probability distribution $\rho^{(1)}$ is defined as:

$$\rho^{(1)}(p, t) = \frac{1}{N} \left\langle \sum_{i=1}^{N} \delta(p - p_i) \right\rangle_{P(\mathbf{p}, t)}, \tag{8}$$

$$= \int_{[0,1]^{N-1}} \mathrm{d}p_2 \mathrm{d}p_3 \ldots \mathrm{d}p_N\, P(\mathbf{p}, t) = P^{(1)}(p, t), \tag{9}$$

and agrees with the marginal probability distribution for the production degree of the first individual $P^{(1)}$. The equality between the normalized reduced one-particle distribution $\rho^{(1)}$ and the one-particle probability distribution $P^{(1)}$ follows from the symmetry of $P$ with respect to permutation of identical (that is indistinguishable) individuals (**Kadar, 2007**).

We also define the more general reduced $n$-particle probability distribution:

$$\rho^{(n)}(p_1,\ldots,p_n,t) := \frac{(N-n)!}{N!}\left\langle \sum_{i_1=1}^{N}\delta(p_1-p_{i_1})\ldots\sum_{\substack{i_n=1\\i_n\neq i_2,\ldots,i_{n-1}}}^{N}\delta(p_n-p_{i_n})\right\rangle_{P(\mathbf{p},t)}, \tag{10}$$

$$= \int\limits_{[0,1]^{N-n}} dp_{n+1}dp_{n+2}\ldots dp_N\, P(\mathbf{p},t) = P^{(n)}(p_1,\cdots,p_n,t), \tag{11}$$

which agrees with the marginal probability distribution for the production degrees of the first $n$ individuals, $P^{(n)}$. In particular, one also has $P(\mathbf{p},t) = P^{(N)}(\mathbf{p},t) = \rho^{(N)}(\mathbf{p},t)$.

## Towards the macroscopic dynamics: Temporal evolution of the reduced one-particle probability distribution

In the following we show that the temporal evolution equation of the reduced one-particle probability distribution is obtained from the master equation **(3)** as:

$$\partial_t \rho^{(1)}(p,t) = 2\lambda \int\limits_{[0,1]^N} dp_1 dp_2\ldots dp_N\, \rho^{(N)}(\mathbf{p},t)(1-sp)\delta(p-R(\langle p\rangle)) \tag{12}$$

$$-2\lambda\left(\rho^{(1)}(p,t)-s\int_0^1 dp_2\,\rho^{(2)}(p,p_2,t)\,p_2\right)$$

$$+(1-2\lambda)s\left(\int_0^1 dp_2\,\rho^{(2)}(p,p_2,t)\,p_2 - p\rho^{(1)}(p,t)\right).$$

To derive the temporal evolution equation for $\rho^{(1)}$, we specify the production degree of one particular individual (here $p_1$), and integrate out the production degrees of the other $N-1$ individuals in the master equation **(3)**:

$$\partial_t\rho^{(1)}(p_1,t) = \int\limits_{[0,1]^{N-1}} dp_2\ldots dp_N\sum_{i=1}^{N}\sum_{j\neq i}^{N}\int\limits_{[0,1]^2}d\widetilde{p}_i d\widetilde{p}_j\, P(\widetilde{\mathbf{p}}^{i,j},t)\frac{\phi(\widetilde{p}_i)}{N-1}A_i(\widetilde{\mathbf{p}}^{i,j};i)A_j(\widetilde{\mathbf{p}}^{i,j};i) \tag{13}$$

$$-\int\limits_{[0,1]^{N-1}} dp_2\ldots dp_N\, P(\mathbf{p},t)\sum_{i=1}^{N}\sum_{j\neq i}^{N}\frac{\phi(p_i)}{N-1},$$

$$= \int\limits_{[0,1]^{N-1}} dp_2\ldots dp_N\,(\text{gain}-\text{loss}) =: I_{\text{gain}} - I_{\text{loss}}.$$

For the loss term, we split the sum $\sum_i{}^{\star}{}_{(i)}$ into two contributions:

$$\sum_i{}^{\star}{}_{(i)} = {}^{\star}{}_{(i=1)} + \sum_{i>1}{}^{\star}{}_{(i)}, \tag{14}$$

and deal with both contributions separately to obtain:

$$I_{\text{loss}} = NP^{(1)}(p,t) - spP^{(1)}(p,t) - s(N-1)\int_0^1 dp_2\, P^{(2)}(p,p_2,t)\,p_2. \tag{15}$$

For the gain term, we split up the sum $\sum_i\sum_{j\neq i}{}^{\star}{}_{(i,j)}$ that occurs in the master equation **(3)** into three terms as follows:

$$\sum_{i\geq 1}\sum_{\substack{j\geq 1\\ j\neq i}}{}^*_{(i,j)} = \sum_{j>1}{}^*_{(i=1,j)} + \sum_{i>1}{}^*_{(i,j=1)} + \sum_{i>1}\sum_{\substack{j>1\\ j\neq i}}{}^*_{(i,j)} \, . \tag{16}$$

We also introduce the notation $\mathrm{d}\widetilde{\mathbf{p}}^{i,j,\hat{k}} := \mathrm{d}p_1 \mathrm{d}p_2 \ldots \mathrm{d}\widetilde{p}_i \ldots \mathrm{d}\widetilde{p}_j \ldots \mathrm{d}\hat{p}_k \ldots \mathrm{d}p_N$ in which variables in the superscript are labeled with a tilde in the product (indices $i$ and $j$ in the example), and variables with a hat in the superscript are missing in the product (that is, they are not integrated over; index $k$ in the example). This way, the integral measure in the gain term can be decomposed as follows:

$$\int_{[0,1]^{N-1}} \mathrm{d}p_2 \ldots \mathrm{d}p_N \sum_{i=1}^{N}\sum_{j\neq i}^{N} \int_{[0,1]^2} \mathrm{d}\widetilde{p}_i \mathrm{d}\widetilde{p}_j$$

$$= \int_{[0,1]^{N+1}} \sum_{j=2}^{N} \mathrm{d}\widetilde{\mathbf{p}}^{1,j}\mathrm{d}p_j + \int_{[0,1]^{N+1}} \sum_{i=2}^{N} \mathrm{d}\widetilde{\mathbf{p}}^{1,i}\mathrm{d}p_i + \int_{[0,1]^{N+1}} \sum_{i=2}^{N}\sum_{\substack{j=2\\ j\neq i}}^{N} \mathrm{d}\mathbf{p}^{\hat{1},i,j}\mathrm{d}p_i \mathrm{d}p_j \, . \tag{17}$$

Upon plugging in the specific form of the transition probabilities and decomposing the integral measure into the three contributions, the gain term can be written as follows (note the asymmetry between the first summand ($i = 1$ term) and the second summand ($j = 1$ term); integration over suitable $\delta$-functions of the transition probabilities was carried out as well, for example, $\int_0^1 \mathrm{d}p_j \, A_j(\widetilde{\mathbf{p}}^{i,j};i) = 1$):

$$\begin{aligned}
I_{\mathrm{gain}} = {} & \frac{1}{N-1}\int_{[0,1]^N} \sum_{j=2}^{N} \mathrm{d}\widetilde{\mathbf{p}}^{1,j}\, P(\widetilde{\mathbf{p}}^{1,j},t)\phi(\widetilde{p}_1)\big(\lambda\delta(p_1 - R(\langle\widetilde{p}^{1,j}\rangle)) + (1-\lambda)\delta(p_1 - \widetilde{p}_1)\big)\\
& + \frac{1}{N-1}\int_{[0,1]^N} \sum_{i=2}^{N} \mathrm{d}\widetilde{\mathbf{p}}^{i,1}\, P(\widetilde{\mathbf{p}}^{i,1},t)\phi(\widetilde{p}_i)\big(\lambda\delta(p_1 - R(\langle\widetilde{p}^{i,1}\rangle)) + (1-\lambda)\delta(p_1 - \widetilde{p}_i)\big)\\
& + \frac{1}{N-1}\int_{[0,1]^{N-1}} \sum_{i=2}^{N}\sum_{\substack{j=2\\ j\neq i}}^{N} \mathrm{d}\widetilde{\mathbf{p}}^{\hat{1},i,j}\, P(\widetilde{\mathbf{p}}^{i,j},t)\phi(\widetilde{p}_i) \, .
\end{aligned} \tag{18}$$

Making use of the fact that $P$ is symmetric with respect to permutation of individuals (individuals are identical), carrying out possible integrals over $\delta$-functions, plugging in the explicit form of the fitness function **(4)**, and relabeling variables, one obtains for the gain term:

$$\begin{aligned}
I_{\mathrm{gain}} = {} & 2\lambda\int_{[0,1]^N} \mathrm{d}\mathbf{p}\, P(\mathbf{p},t)\delta(p - R(\langle p\rangle))(1-sp_1)\\
& + 2(1-\lambda)(1-sp)P^{(1)}(p,t)\\
& + (N-2)P^{(1)}(p,t) - s(N-2)\int_0^1 \mathrm{d}p_2 \, P^{(2)}(p,p_2,t)\, p_2 \, .
\end{aligned} \tag{19}$$

Combining loss terms $I_{\mathrm{loss}}$ and gain terms $I_{\mathrm{gain}}$ leads to the result for the equation of motion of the reduced one-particle probability distribution $\rho^{(1)}$ that is given in **Equation (12)**.

## Heuristic derivation of the macroscopic dynamics: Mean-field approximation

Upon assuming that correlations are negligible, one may approximate $\rho^{(1)}$ by its mean-field approximation $\rho$, which we refer to as the production distribution. As described in the main text, the temporal evolution equation for $\rho^{(1)}$ serves as a suitable starting point to guess the

mean-field equation for $\rho$, which is the mean-field approximation of $\rho^{(1)}$. Thus, we naively approximate $\rho^{(1)} \approx \rho$ and $\rho^{(N)} \approx \prod^N \rho$. From the temporal evolution of $\rho^{(1)}$ in **Equation (12)**, the mean-field equation for $\rho$ is suggested as:

$$
\begin{aligned}
\partial_t \rho(p,t) \;\approx\; & 2\lambda \int_{[0,1]^N} \mathrm{d}p_1 \mathrm{d}p_2 \ldots \mathrm{d}p_N \prod_{i=1}^{N} \rho(p_i)\, (1 - s\overline{p}_t)\delta(p - R(\overline{p}_t)) \\
& - 2\lambda \left( \rho(p,t) - s \int_0^1 \mathrm{d}p_2\, \rho(p,t)\rho(p_2,t)p_2 \right) \\
& + (1 - 2\lambda)s \left( \int_0^1 \mathrm{d}p_2\, \rho(p,t)\rho(p_2,t)p_2 - p\rho(p,t) \right),
\end{aligned}
\tag{20}
$$

where $\overline{\cdot}_t$ denotes averaging with respect to $\rho$ at time $t$. Further collection of terms yields the mean-field equation **(1)** in the main text:

$$
\partial_t \rho(p,t) = 2\lambda \overline{\phi}_t \big( \delta(p - R(\overline{p}_t)) - \rho(p,t) \big) + (1 - 2\lambda)\big( \phi(p) - \overline{\phi}_t \big)\rho(p,t),
\tag{21}
$$

with initial condition $\rho(p, t = 0) = \rho_0(p)$, $\phi(p) = 1 - sp$, $\overline{\phi}_t = 1 - s\overline{p}_t$, and $\overline{p}_t = \int_0^1 \mathrm{d}p\, p\rho(p,t)$. Alternatively, this mean-field equation can also be written as:

$$
\partial_t \rho(p,t) = 2\lambda \big( \overline{\phi}_t \delta(p - R(\overline{p}_t)) - \phi(p)\rho(p,t) \big) + \big( \phi(p) - \overline{\phi}_t \big)\rho(p,t).
\tag{22}
$$

We emphasize that the mean-field equation **(1)** is to be understood in distributional sense, that is, it needs to be integrated over observables (for example, suitable test functions $g : [0,1] \to \mathbb{R}$, $g$ smooth) and $\rho$ is interpreted as a linear functional on the space of these observables. This way, $\rho$ can be a continuous probability density function or a discrete probability mass function, or a probability distribution with both density parts and mass parts. To keep notation accessible for a broad readership, we avoid a measure-theoretic notation in this manuscript.

The proof that $\rho^{(1)}$ converges in probability to $\rho$ as $N \to \infty$ for any finite time if initial correlations are not too strong will be presented in a forthcoming publication (**Frey et al., 2017**).

## Appendix 3

# Analysis of the mean-field equation of the quorum-sensing model (autoinducer equation)

## Mean-field equation for moment and cumulant-generating functions

The moment-generating function $M(u,t)$ for the production degree $p$, which is the random variable of interest, and its corresponding cumulant-generating function $C(u,t)$ are defined as:

$$M(u,t) := \int_0^1 dp\, e^{up} \rho(p,t) = \mathcal{L}[\rho](-u,t)\,, \tag{23}$$

$$C(u,t) := \ln(M(u,t))\,, \tag{24}$$

with argument $u \in (-\infty,\infty)$ at time $t$. The moment-generating function $M$ is the (one-sided) Laplace transform $\mathcal{L}$ of $\rho$ with negative argument at time $t$. Moments and cumulants of the degree distribution $\rho$ are obtained as:

$$M_k(t) := \partial_u^k M(u,t)|_{u=0}\,, \text{ and } C_k(t) := \partial_u^k C(u,t)|_{u=0}\,, \text{ for } k \geq 1\,. \tag{25}$$

For the mean production, that is, for the expectation value of the production distribution, it holds that $\bar{p} = M_1 = C_1$ and the variance is given by $\mathrm{Var}(p) = \overline{p^2} - \bar{p}^2 = M_2 - M_1^2 = C_2$. By applying transformations (**23, 24**) to the mean-field equation **(1)** and plugging in the form of the fitness function in **Equation (4)**, one obtains:

$$\begin{aligned}
\partial_t M(u,t) &= (1-2\lambda)s(M_1(t)M(u,t) - \partial_u M(u,t)) + 2\lambda(1-sM_1(t))\left(e^{uR(M_1(t))} - M(u,t)\right), \\
\partial_t C(u,t) &= (1-2\lambda)s(C_1(t) - \partial_u C(u,t)) + 2\lambda(1-sC_1(t))\left(e^{uR(C_1(t))}e^{-C(u,t)} - 1\right).
\end{aligned} \tag{26}$$

## Solution strategy for the moment and cumulant-generating functions: Method of characteristics

This mean-field equation in moment/cumulant space **(26)** is more conveniently written as a semilinear partial differential equation (PDE) of first order in $t$ and $u$, for example for $C$:

$$\partial_t C(u,t) + (1-2\lambda)s\partial_u C(u,t) = F(C,u,t)\,, \tag{27}$$

with $F(C,u,t) := (1-2\lambda)sC_1(t) + 2\lambda(1-sC_1)\left(e^{uR(C_1(t))}e^{-C(u,t)} - 1\right)$ and initial condition $C(u,t=0) = C_0(u)$. This PDE admits the straight lines $r(u,t) = u - (1-2\lambda)st$ as characteristics. Restricted to these characteristic curves, the PDE reduces to a nonlinear ordinary differential equation (ODE) of first order in time for $z(r,t) = C(u(r,t),t)$:

$$\frac{d}{dt}z(r,t) = \partial_t u(r,t)\partial_u C(u,t) + \partial_t C(u,t) = F(C(u(r,t),t),u(r,t),t) = F(z,r,t)\,, \tag{28}$$

with initial condition $z(r,t=0) = C(u(r,t=0),t=0) = C_0(r)$. The solution for the cumulant-generating function is then obtained from the solution of the above ODE as $C(u,t) = z(r(u,t),t) = z(u - (1-2\lambda)st,t)$. For the two cases $\lambda = 0$ and $\lambda = 1/2$ with linear response function, an insightful, analytical solution of the mean-field equation **(1)** for the production distribution for all times $t$ was found this way; see below.

## Moment and cumulant equations

A different approach to characterize the dynamics of the quorum-sensing model is to analyze the equations of motions for the moments and cumulants. The moment equations are derived from *Equation (26)* by applying the definition of the moments *(25)*, which yields for $k \geq 1$,

$$\partial_t M_k(t) = (1 - 2\lambda)s(M_1(t)M_k(t) - M_{k+1}(t)) + 2\lambda(1 - sM_1(t))\big(R^k(M_1(t)) - M_k(t)\big). \tag{29}$$

The equations for the first three cumulants are obtained as,

$$
\begin{aligned}
\partial_t C_1(t) &= -(1 - 2\lambda)sC_2(t) + 2\lambda(1 - sC_1(t))(R(C_1(t)) - C_1(t)) , \\
\partial_t C_2(t) &= -(1 - 2\lambda)sC_3(t) + 2\lambda(1 - sC_1(t))\Big(-C_2(t) + (R(C_1(t)) - C_1(t))^2\Big) , \\
\partial_t C_3(t) &= -(1 - 2\lambda)sC_4(t) + 2\lambda(1 - sC_1(t))\Big(-C_3(t) \\
&\qquad -3(R(C_1(t)) - C_1(t))C_2(t) + (R(C_1(t)) - C_1(t))^3\Big) .
\end{aligned}
\tag{30}
$$

For *Figure 2E* of the main text, the cumulant equations *(30)* were numerically integrated after applying a Gaussian approximation, that is a cumulant closure with $C_i(t) = 0$ for $i \geq 3$ and all $t$, and plotted for $\bar{p}_t = C_1(t)$.

## Without sense-and-response ($\lambda = 0$): Analytical solution and approach of the homogeneous stationary distribution of non-producers

For the case without sense-and-response through quorum sensing, $\lambda = 0$, it is readily seen from *Equation (1)* that stationary production distributions are given by $\delta$-peaks as $\rho_\infty(p) := \rho(p, t \to \infty) = \delta(p - p_{\text{low}})$ for all $p_{\text{low}} \in [0, 1]$. However, the distribution with solely non-producers, $p_{\text{low}} = 0$, is the only asymptotically stable solution of the mean-field equation *(1)*; see below.

When sense-and-response is absent, the analytical solution of the mean-field equation *(1)* for $\rho$ can be obtained by applying the method of characteristics to *Equation (27)* as outlined above. The implicit solution is given by:

$$C(u, t) = C_0(u - st) + st\langle\bar{p}\rangle_t , \quad \text{with} \quad \langle\bar{p}\rangle_t := 1/t \int_0^t \mathrm{d}t' \, \bar{p}_{t'} \tag{31}$$

as the temporal average of the mean production $\bar{p}_t$. Back-transformation and exploiting normalization of $\rho$ yields:

$$\rho(p, t) = \rho_0(p)e^{-st(p - \langle\bar{p}\rangle_t)} = \rho_0(p)e^{-stp}/\mathcal{L}[\rho_0](st) . \tag{32}$$

For example, if the initial production distribution $\rho_0$ is a uniform distribution on $[0, 1]$, $\rho$ evolves in time as $\rho(p, t) = st/(1 - e^{-st})e^{-stp}$, which is plotted in *Figure 2A* of the main text (black, solid lines). Every production degree that is different from $p = 0$ decays exponentially fast and the time scale of the decay is set by the inverse of the value of that production degree. As $p \to 0$, this time scale diverges and, hence, the stationary distribution,

$$\rho_\infty(p) = \delta(p) , \tag{33}$$

is approached algebraically slowly; see *Figure 2D* of the main text.

To quantify the dependence of the time scales to approach stationarity on the initial distribution in more generality, we analyzed the temporal solution of the mean $\bar{p}_t$, which is obtained from the solution for the cumulant generating function as:

$$\overline{p}_t = -\partial_v \ln \mathcal{L}[\rho_0](v)|_{v=st} \, . \tag{34}$$

Therefore, the temporal evolution of the mean production depends only on the initial distribution $\rho_0$ via its Laplace transform $\mathcal{L}[\rho_0]$. For the asymptotic behavior of Laplace transforms it is known that if $\rho_0(p) \sim p^\mu$ as $p \to 0$ with $\mu > -1$, then $\mathcal{L}[\rho_0](v) \sim 1/v^{(\mu+1)}$ for $v \gg 1$ (**Doetsch, 1976**). Therefore, it follows that the mean evolves in time as $\overline{p}_t \sim 1/t$ for $t \gg 1$ if the initial production distribution is a continuous probability density with non-vanishing weight at $p_{\text{low}} = 0$ (chosen for simplicity as the lowest production degree). The condition that the exponent satisfies $\mu > -1$ is always fulfilled for a continuous probability distribution to ensure integrability at zero. In the same manner, the decay of the variance is shown to evolve in time algebraically as $\text{Var}(p)(t) \sim 1/t^2$ for $t \gg 1$.

In contrast, if the lowest production degree is separated from all other degrees in the population by a gap $\Delta > 0$ in production space, mean and variance approach their stationary value exponentially fast at a time scale set by $\Delta$. To see this qualitative difference in the approach of stationarity, we consider an initial probability distribution with probability mass $y_0 > 0$ at degree $p_{\text{low}} = 0$ (chosen again for simplicity) and a remainder probability distribution $\widetilde{\rho}_0$ with support on $[\Delta, 1]$: $\rho_0(p) = y_0 \delta(p) + (1-y_0)\widetilde{\rho}_0(p)\mathbb{I}_{[\Delta,1]}(p)$ (here $\mathbb{I}_{[\Delta,1]}$ denotes the indicator function, which takes value 1 on the interval $[\Delta, 1]$ and 0 otherwise, and highlights the support of $\widetilde{\rho}_0$ on $[\Delta, 1]$). Using this form for $\rho_0$ and plugging in its Laplace transform into the solution for the mean in **Equation (34)**, one estimates $\overline{p}_t \lesssim (1+\Delta)e^{-s\Delta \cdot t}$ for $t \gg 1$. This result generalizes the exponentially fast approach of stationarity that is known, for example, from the discrete Prisoner's dilemma in evolutionary game theory (**Nowak et al., 2004**; **Traulsen et al., 2005**; **Melbinger et al., 2010**; **Assaf et al., 2013**).

In total, $\overline{p}_t$ vanishes exponentially fast if and only if the production degree at the smallest production degree is separated by a gap $\Delta$ from all other production degrees that are present in the population. On the other hand, if the lowest production degree is part of an interval with continuously distributed production degrees (that is, $\Delta = 0$), $\overline{p}_t$ decreases algebraically slowly.

## With sense-and-response ($\lambda > 0$): Homogeneous stationary distributions

For the case with sense-and-response through quorum sensing, $\lambda > 0$, one obtains from **Equation (1)** or from the cumulant equations **(30)** that stationary production distributions are given by $\delta$-peaks as:

$$\rho_\infty(p) = \delta(p - p^*) \, , \quad \text{with} \ \ R(p^*) = p^* \in [0, 1] \, . \tag{35}$$

In other words, fixed points of the response function give rise to homogeneous stationary distributions. Whether these stationary distributions are stable against small perturbations around stationarity depends on the stability of the fixed points (see linear stability analysis of homogeneous stationary distributions below). Whether they are approached for long times does not only depend on the stability of the fixed points, but also on the initial distribution, the response function, and the value of $\lambda$ (see heterogeneous stationary distributions).

## Linear stability analysis of homogeneous stationary distributions

Here, we supplement the statements from the main text on the stability of homogeneous stationary distributions in the linear approximation around stationarity if sense-and-response is present ($\lambda > 0$). For the sake of simplicity and feasibility, we carry out the stability analysis in the space of cumulants. To this end, we define the vector:

$$\mathbf{C}(t) = (C_1(t), C_2(t), C_3(t), \dots) \,, \tag{36}$$

which is at stationarity (see **Equation (35)**):

$$\mathbf{C}(t \to \infty) = \mathbf{C}_\infty = (C_{1,\infty}, C_{2,\infty}, C_{3,\infty}, \dots) = (p^*, 0, 0, \dots) \,. \tag{37}$$

With this notation, the equations of motion for the cumulants of $\rho$ are given as follows:

$$\partial_t C_i(t) = F_i(\mathbf{C}(t)) \,, \quad \text{for } i \geq 1 \,. \tag{38}$$

Here, the functions $F_i$ for $i \geq 0$ are defined by the right hand side of the cumulant equations **(30)**. Upon introducing the distance $\Delta \mathbf{C}$ to the stationary vector $\mathbf{C}_\infty$, that is $\Delta \mathbf{C} = \mathbf{C} - \mathbf{C}_\infty$, one obtains the temporal behavior of $\Delta \mathbf{C}$ as:

$$\partial_t \Delta C_i(t) = F_i(\mathbf{C}_\infty + \Delta \mathbf{C}(t)) = \sum_{j=0}^{\infty} J_{ij}(\mathbf{C}_\infty) \Delta C_j(t) + \mathcal{O}(\|\Delta \mathbf{C}\|^2) \,, \quad \text{for } i \geq 0, \tag{39}$$

with Jacobian $J_{ij}(\mathbf{C}_\infty) = \frac{\partial F_i(\mathbf{C})}{\partial C_j}\big|_{\mathbf{C}=\mathbf{C}_\infty}$, whose entries are obtained after some algebra as:

$$J_{11} = -2\lambda(1 - sp^*)(1 - R'(p^*)) \,, \tag{40}$$
$$J_{i,i} = -2\lambda(1 - sp^*) \,, \quad \text{for } i \geq 2 \,, \tag{41}$$
$$J_{i,i+1} = -(1 - 2\lambda)s \,, \quad \text{for } i \geq 1 \,, \tag{42}$$
$$J_{i,j} = 0 \,, \quad \text{otherwise} \,. \tag{43}$$

The eigenvalues of the upper triangular matrix $J$ determine the stability of the stationary distribution up to linear order in perturbations at the level of cumulants around stationarity. Because of the upper triangular structure of the Jacobian $J$, its eigenvalues are given by the diagonal entries of $J$:

$$\gamma_1 = -2\lambda(1 - sp^*)(1 - R'(p^*)) \,, \tag{44}$$
$$\gamma_i = -(1 - 2\lambda)s < 0 \,, \quad \text{for } i \geq 2 \,. \tag{45}$$

Thus, local stability of homogeneous stationary distributions ($\rho_\infty(p) = \delta(p - p^*)$ with $R(p^*) = p^*$) is determined by the stability of the fixed points, that is whether $R'(p^*)$ is less or greater than 1.

In total, homogeneous stationary distributions are unstable up to linear order in perturbations at the level of cumulants around stationarity if $R'(p^*) > 1$. In other words, stationary distributions located at a fixed point $p^*$ are linearly unstable if $p^*$ is an unstable fixed point of the response function ($R'(p^*) > 1$). On the other hand, linear stability of the response function at $p^*$ ($R'(p^*) \leq 1$) yields linearly stable homogeneous stationary distributions located at $p^*$.

## With sense-and-response ($\lambda = 1/2$) and linear response function ($R(p) = p$): Analytical solution and approach of homogeneous stationary distribution

For the choice of linear response function ($R(p) = p$, that is, $R'(p) = 1$ for all $p \in [0, 1]$) and $\lambda = 1/2$, the mean remains constant in time (see **Equation (30)**). Furthermore, one obtains the analytical solution of the mean-field equation **(1)** by applying the method of characteristics (most conveniently in the space of moment generating functions) as:

$$M(u,t) = M_0(u)e^{-\overline{\phi}_0 t} + e^{u\overline{p}_0}\left(1 - e^{-\overline{\phi}_0 t}\right) , \qquad (46)$$

which yields after back-transformation:

$$\rho(p,t) = y(t)\rho_0(p) + (1 - y(t))\delta(p - \overline{p}_0) , \quad \text{with } y(t) = \exp(-\overline{\phi}_0 t) . \qquad (47)$$

The initial production distribution $\rho_0$ decays exponentially fast on a time scale that is set by the average initial fitness in the population $\overline{\phi}_0$, whereas a singular probability mass at the initial mean production degree $\overline{p}_0$ builds up concomitantly due to sense-and-response through quorum sensing. The population approaches the stationary distribution $\rho_\infty(p) = \delta(p - \overline{p}_0)$ exponentially fast.

## With sense-and-response ($\lambda = 1/2$) and polynomial response function: Divergence of time scales at bifurcations of parameters of the response function

For $\lambda > s/2$, the approach of stationarity is typically exponentially fast. However, upon fine-tuning parameters of the response function one observes an algebraically slow approach of stationarity. We exemplify this qualitative change in the temporal evolution by setting the response probability to $\lambda = 1/2$ and by considering the following nonlinear response function, see **Appendix 1—figure 3** (for the sake of readability, we label the argument of $R$ by $p$ instead of $\langle p \rangle$):

$$R(p) = p + A \cdot p(p - (p_{cr} - \epsilon))(p - p_{cr})(p - (p_{cr} + \epsilon))(p - 1) , \qquad (48)$$

with some real constant $A > 0$. The chosen response function **(48)** is a polynomial of fifth order with $R(0) = 0$ and $R(1) = 1$, and parameter $0 < p_{cr} < 1$, which is set to $p_{cr} = 1/2$ in **Appendix 1— figure 3**. The bifurcation parameter $0 \leq \epsilon \leq \min(p_{cr}, 1 - p_{cr})$ controls a supercritical pitchfork bifurcation of the response function **(48)** at $p^* = p_{cr}$: Whereas $p^* = 0$ and $p^* = 1$ are unstable fixed points for all $\epsilon$, the fixed points at $p^* = p_{cr} \pm \epsilon$ are stable for $\epsilon > 0$ and merge with $p^* = p_{cr}$ for $\epsilon = 0$. The fixed point $p^* = p_{cr}$ is unstable for $\epsilon > 0$ and is a three-fold degenerate, stable fixed point for $\epsilon = 0$, see **Appendix 1—figure 3A,B**.

For $\lambda = 1/2$ and upon plugging in the explicit form of the response function **(48)**, the temporal evolution equation of the mean **(30)** is given by the ODE:

$$\partial_t C_1 = A(1 - sC_1)C_1(C_1 - (p_{cr} - \epsilon))(C_1 - p_{cr})(C_1 - (p_{cr} + \epsilon))(C_1 - 1) , \qquad (49)$$

with initial condition $C_1(t = 0) = \overline{p}_0$. From integrating this temporal evolution equation, one obtains the implicit solution for the mean $\overline{p} = C_1$ as:

$$t = \sum_{p^*} \alpha_{p^*} \int_{\overline{p}_0}^{\overline{p}_t} \frac{dC_1}{C_1 - p^*} . \qquad (50)$$

The sum is performed over all non-degenerate fixed points of the right hand side of the equation for the mean **(49)**, that is over the roots $p^* \in \{0, p_{cr} - \epsilon, p_{cr}, p_{cr} + \epsilon, 1, 1/s\}$ of both the response function **(48)** and the mean fitness $\overline{\phi}_t = 1 - s\overline{p}_t$. The coefficients $\alpha_{p^*}$ arise from the partial fraction decomposition with $\alpha_{p_{cr}, p_{cr} \pm \epsilon} \sim \mathcal{O}(1/\epsilon^2)$ and $\alpha_{0,1,1/s} \sim \mathcal{O}(\epsilon^0)$. Therefore, one concludes that:

$$|\overline{p}_t - \overline{p}_\infty| \sim e^{-t/\alpha} , \ \text{ for } \epsilon > 0 , \tag{51}$$

for large times and with a decay constant $\alpha$ that diverges as the bifurcation is approached as $\alpha \sim 1/\epsilon^2$. In other words, stationarity is approached exponentially fast when all fixed points of the response function **(48)** are non-degenerate, see **Appendix 1—figure 3D** inset. Which of the two stable fixed points $p^* = p_{\mathrm{cr}} \pm \epsilon$ constitutes the stationary distribution $\rho_\infty(p) = \delta(p - p^*)$ depends on the initial distribution (and demographic fluctuations of the initial dynamics in the stochastic process). The prediction that the decay constant $\tau$ diverges as the bifurcation of the response function is approached ($\epsilon \rightarrow 0$) is in good agreement with numerical simulations of the stochastic process, see **Appendix 1—figure 3D**.

In contrast to the exponentially fast approach away from the bifurcation, stationarity is approached algebraically slowly at the bifurcation of the nonlinear response function, that is, for $\epsilon = 0$. Since the stable fixed point $p^* = p_{\mathrm{cr}}$ is three-fold degenerate, one finds by integration of **Equation (50)** the implicit solution for the mean as:

$$t = \sum_{p^* \neq p_{\mathrm{cr}}} \alpha_{p^*} \int_{\overline{p}_0}^{\overline{p}_t} \frac{\mathrm{d}C_1}{C_1 - p^*} + \sum_{i=1}^{z} \alpha_{p_{\mathrm{cr}}}^{(i)} \int_{\overline{p}_0}^{\overline{p}_t} \frac{\mathrm{d}C_1}{(C_1 - p_{\mathrm{cr}})^i} . \tag{52}$$

In addition to the sum over the non-degenerate fixed points ($p^* \neq p_{\mathrm{cr}}$), a second sum accounts for the degeneracy $z = 3$ of the fixed point $p_{\mathrm{cr}}$, which is reflected by the singularities in the integrand up to order $z$. Consequently, the mean production approaches its stationary value as:

$$|\overline{p}_t - \overline{p}_\infty| \sim t^{-1/\nu} , \ \text{ for } \epsilon = 0 , \tag{53}$$

for large times with critical exponent $\nu = z - 1 = 2$, that is $-1/\nu = -1/2$. **Appendix 1—figure 3C** shows the excellent agreement of our theoretical predictions with numerical simulations of the stochastic process for the algebraically slow approach of stationarity at the bifurcation.

## With rare sense-and-response ($0 < \lambda < s/2$): Heterogeneous stationary distributions

To analyze heterogeneous stationary distributions, we decompose the production distribution as follows:

$$\rho(p,t) = y(t)\rho_{\mathrm{low}}(p,t) + (1 - y(t))\rho_{\mathrm{high}}(p,t) \tag{54}$$

where $\rho_{\mathrm{low}}$ and $\rho_{\mathrm{high}}$ denote two probability distributions with support on the interval $[0,1]$. Their respective means are denoted as:

$$\overline{p}_{\mathrm{low},t} = \int_0^1 \mathrm{d}p \, p\rho_{\mathrm{low}}(p,t) , \ \text{ and } \ \overline{p}_{\mathrm{high},t} = \int_0^1 \mathrm{d}p \, p\rho_{\mathrm{high}}(p,t) , \tag{55}$$

such that $\overline{p}_t = y(t)\overline{p}_{\mathrm{low},t} + (1 - y(t))\overline{p}_{\mathrm{high},t}$; their stationary values are denoted as $\overline{p}_{\mathrm{low},\infty} =: p_{\mathrm{low}}$ and $\overline{p}_{\mathrm{high},\infty} =: p_{\mathrm{high}}$, respectively. We decompose the initial distribution $\rho_0(p) = y_0\rho_{\mathrm{low},0}(p) + (1 - y_0)\rho_{\mathrm{high},0}(p)$ such that $\min(\mathrm{supp}(\rho_{\mathrm{low},0})) = \min(\mathrm{supp}(\rho_0))$. For a numerical integration of the mean-field equation **(1)** that not only reproduces the stationary distribution, but also the temporal approach towards stationarity, it turns out suitable to choose the following decomposition: $\rho_{\mathrm{low},0} = \rho_0$, $\rho_{\mathrm{high},0} = \delta(\cdot - R(\overline{p}_0))$, and $y_0 = 1 - \epsilon$ with $0 < \epsilon \leq 0.01$.

With decomposition **(54)**, the mean-field equation **(1)** for $\rho$ can be rewritten in terms of equations for $\rho_{\mathrm{low}}, \rho_{\mathrm{high}}$, and $y$ as follows:

$$\partial_t \rho_{\text{low}}(p,t) = -s(1-2\lambda)\left(p-\overline{p}_{\text{low},t}\right)\rho_{\text{low}}(p,t) \,, \tag{56}$$

$$\partial_t \rho_{\text{high}}(p,t) = -s(1-2\lambda)\left(p-\overline{p}_{\text{high},t}\right)\rho_{\text{high}}(p,t) + 2\lambda\frac{1-s\overline{p}_t}{1-y(t)}\left(\delta(p-R(\overline{p}_t)) - \rho_{\text{high}}(p,t)\right) \,, \tag{57}$$

$$\partial_t y(t) = y(t)\left(-2\lambda(1-s\overline{p}_{\text{low},t}) + s(1-y(t))(\overline{p}_{\text{high},t} - \overline{p}_{\text{low},t})\right) \,. \tag{58}$$

We note that the decomposition *(54)* of $\rho$ with *Equations (56–58)* is not unique, but this choice of decomposition enables the characterization of heterogeneous stationary distributions and, thus, phenotypic heterogeneity.

The temporal evolution equation *(56)* for $\rho_{\text{low}}$ has the form of the continuous replicator equation (see *Equation (1)* with $\lambda = 0$) with renormalized selection strength $s(1-2\lambda)$. Following the analysis that resulted in *Equation (32)*, the solution for $\rho_{\text{low}}$ is given by:

$$\rho_{\text{low}}(p,t) = \rho_{\text{low},0}(p)e^{-s(1-2\lambda)tp}/\mathcal{L}[\rho_{\text{low},0}](s(1-2\lambda)t) \,, \quad \text{with} \ \ \rho_{\text{low},0}(p) = \rho_{\text{low}}(p,t=0) \,, \tag{59}$$

if $\lambda \leq 1/2$. As shown in the main text, the condition $\lambda \leq 1/2$ is consistent with the condition for the upper threshold of the response probability $\lambda \leq s/2 < 1/2$, above which heterogeneous stationary distributions cannot occur. For the mean $\overline{p}_{\text{low},t}$, one obtains:

$$\overline{p}_{\text{low},t} = -\partial_v \ln \mathcal{L}[\rho_{\text{low},0}](v)\big|_{v=s(1-2\lambda)t} \,. \tag{60}$$

In other words, $\rho_{\text{low}}$ approaches a stationary $\delta$-distribution:

$$\rho_{\text{low}}(p,t\to\infty) = \rho_{\text{low},\infty}(p) = \delta(p - p_{\text{low}}) \,,$$
$$\text{with} \ \ p_{\text{low}} = \overline{p}_{\text{low},\infty} = \min(\text{supp}(\rho_{\text{low},0})) = \min(\text{supp}(\rho_0)) \,. \tag{61}$$

The temporal evolution equation *(57)* for $\rho_{\text{high}}$ has a similar form as the original mean-field equation *(1)*: it involves the sense-and-response term with prefactor $2\lambda$, and the replicator term with prefactor $1-2\lambda$. The sense-and-response term, however, couples to the full production distribution $\rho$ through the argument $R(\overline{p}_t)$ in the $\delta$-function and the prefactor $(1-s\overline{p}_t)/(1-y(t))$, whereas the replicator term does not couple to $\rho_{\text{low}}$ or $y$. *Equation (57)* is most suitably analyzed in the space of moment and cumulant generating functions with:

$$M_{\text{high}}(u,t) := \int_0^1 \mathrm{d}p \ e^{up}\rho_{\text{high}}(p,t) \,, \quad \text{and} \ \ C_{\text{high}}(u,t) := \ln\left(M_{\text{high}}(u,t)\right) \,, \quad u \in (-\infty,\infty) \,. \tag{62}$$

The moments and cumulants of $\rho_{\text{high}}$ are obtained as $M_{\text{high},k}(t) := \partial_u^k M_{\text{high}}(u,t)\big|_{u=0}$ and $C_{\text{high},k}(t) := \partial_u^k C_{\text{high}}(u,t)\big|_{u=0}$ for $k \geq 1$. With this notation, it is $\overline{p}_{\text{high},t} = M_{\text{high},1}(t) = C_{\text{high},1}(t)$. By applying these transformations to the temporal evolution equation *(57)* of $\rho_{\text{high}}$, one obtains:

$$\partial_t M_{\text{high}}(u,t) = -(1-2\lambda)s\left(\partial_u M_{\text{high}}(u,t) - M_{\text{high},1}(t)M_{\text{high}}(u,t)\right)$$
$$+ 2\lambda\frac{1-s\overline{p}_t}{1-y(t)}\left(e^{uR(\overline{p}_t)} - M_{\text{high}}(u,t)\right) \,, \tag{63}$$

$$\partial_t C_{\text{high}}(u,t) = -(1-2\lambda)s\left(\partial_u C_{\text{high}}(u,t) - C_{\text{high},1}(t)\right) + 2\lambda\frac{1-s\overline{p}_t}{1-y(t)}\left(e^{uR(\overline{p}_t)}e^{-C_{\text{high}}(u,t)} - 1\right) \,, \tag{64}$$

in which the coupling of $\rho_{\text{high}}$ to $\rho_{\text{low}}$ and $y$ is apparent explicitly through the occurrence of the factor $1-y(t)$ and implicitly through the occurrence of $\overline{p}_t = y(t)\overline{p}_{\text{low},t} + (1-y(t))\overline{p}_{\text{high},t}$. The corresponding equations of motion for the first three cumulants are, thus, obtained as:

$$\partial_t C_{\text{high},1}(t) = -(1-2\lambda)sC_{\text{high},2}(t) + 2\lambda\frac{1-s\bar{p}_t}{1-y(t)}\left(R(\bar{p}_t) - C_{\text{high},1}(t)\right),$$

$$\partial_t C_{\text{high},2}(t) = -(1-2\lambda)sC_{\text{high},3}(t) + 2\lambda\frac{1-s\bar{p}_t}{1-y(t)}\left(-C_{\text{high},2}(t) + (R(\bar{p}_t) - C_{\text{high},1}(t))^2\right),$$

$$\partial_t C_{\text{high},3}(t) = -(1-2\lambda)sC_{\text{high},4}(t) + 2\lambda\frac{1-s\bar{p}_t}{1-y(t)}\Big(-C_{\text{high},3}(t)$$

$$-3(R(\bar{p}_t) - C_{\text{high},1}(t))C_{\text{high},2}(t) + (R(\bar{p}_t) - C_{\text{high},1}(t))^3\Big).$$

(65)

At stationarity, it is $\partial_t y(t) = 0$ and $y(t) \equiv y_\infty$ with (see **Equation (58)**; recall also that $\bar{p}_\infty = y_\infty p_{\text{low}} + (1-y_\infty)p_{\text{high}}$):

$$2\lambda(1-sp_{\text{low}}) = s(1-y_\infty)(p_{\text{high}} - p_{\text{low}}), \text{ or equivalently } (1-2\lambda)(1-sp_{\text{low}}) = 1 - s\bar{p}_\infty. \quad (66)$$

Thus, assuming that a stationary value $0 < y_\infty < 1$ exists, it fulfils the self-consistency relation:

$$y_\infty = 1 - \frac{2\lambda}{s}\frac{1-sp_{\text{low}}}{p_{\text{high}} - p_{\text{low}}} = \frac{p_{\text{high}} - \bar{p}_\infty}{p_{\text{high}} - p_{\text{low}}}. \quad (67)$$

Note that we denoted $y_\infty$ simply as $y$ in the main text.

If $0 < y_\infty < 1$ exists, it follows that the stationary solution for $\rho_{\text{high}}$ can be obtained via **Equation (63)** in terms of the stationary moment generating function $M_{\text{high},\infty}(u) = M_{\text{high}}(u, t \to \infty)$ with:

$$\partial_u M_{\text{high},\infty}(u) - p_{\text{low}}M_{\text{high},\infty}(u) = (p_{\text{high}} - p_{\text{low}})e^{uR(\bar{p}_\infty)}, \text{ and } p_{\text{high}} = \partial_u M_{\text{high},\infty}(u)|_{u=0}, \quad (68)$$

where the relation between $p_{\text{low}}, p_{\text{high}}$, and $y_\infty$ in **Equation (66)** was exploited and the definition $p_{\text{high}} = \bar{p}_{\text{high},\infty}$ translates into the boundary condition. In total, one obtains $M_{\text{high},\infty}(u) = e^{up_{\text{high}}}$ with the self-consistency relation $p_{\text{high}} = R(\bar{p}_\infty)$. In other words, $\rho_{\text{high}}$ approaches a stationary $\delta$-distribution:

$$\rho_{\text{high}}(p, t \to \infty) = \rho_{\text{high},\infty}(p) = \delta(p - p_{\text{high}}),$$

$$\text{with } p_{\text{high}} = \bar{p}_{\text{high},\infty} = R(\bar{p}_\infty) = R(2\lambda/s + (1-2\lambda)p_{\text{low}}). \quad (69)$$

For $p_{\text{low}} = \min(\text{supp}(\rho_0)) = 0$, one recovers from **Equations (61, 67, 69)** the heterogeneous stationary distribution **(2)** that was given in the main text.

For **Figure 2F** of the main text, equations **(58, 60**, and **65)** were numerically integrated with $C_{\text{high},i}(t) = 0$ for $i \geq 3$ and for all $t$, and initial conditions $y_0 = 0.99$, $\rho_{\text{low},0} \sim \text{Uniform}(0, 1)$, and $\rho_{\text{high},0} \sim \delta(\cdot - R(0.5))$. The choice of initial conditions, however, is not important for the asymptotic behavior, see **Appendix 1—figure 1**.

## Linear stability analysis of heterogeneous stationary distributions

Here, we supplement the statements from the main text on the stability of heterogeneous stationary distributions **(2)** in the linear approximation around stationarity. For the sake of simplicity and feasibility, we carry out the stability analysis in the space of cumulants. To this end, we define the vector:

$$\mathbf{c}(t) = (y(t), C_{\text{low},1}(t), C_{\text{high},1}(t), C_{\text{low},2}(t), C_{\text{high},2}(t), \dots) = (c_0(t), c_1(t), c_2(t), \dots), \quad (70)$$

which is at stationarity:

$$\mathbf{c}(t \to \infty) = \mathbf{c}_\infty = (y, C_{\text{low},1}, C_{\text{high},1}, C_{\text{low},2}, C_{\text{high},2}, \ldots) = (y, p_{\text{low}}, p_{\text{high}}, 0, 0, \ldots) = (c_0, c_1, c_2, \ldots) . \quad (71)$$

The cumulants of $\rho_{\text{low}}$ are obtained in the same way as for $\rho_{\text{high}}$, that is as $C_{\text{low},k}(t) := \partial_u^k C_{\text{low}}(u,t)|_{u=0}$ for $k \geq 1$ from $M_{\text{low}}(u,t) := \int_0^1 dp\, e^{up} \rho_{\text{low}}(p,t)$ and $C_{\text{low}}(u,t) := \ln(M_{\text{low}}(u,t))$ for $u \in (-\infty, \infty)$. With this notation, the equations of motion for $y(t)$ in **Equation (58)** and the cumulants of $\rho_{\text{low}}$ and $\rho_{\text{high}}$, respectively, are cast into the compact form:

$$\partial_t c_i(t) = F_i(\mathbf{c}(t)), \quad \text{for } i \geq 0 . \quad (72)$$

Upon introducing the distance $\Delta\mathbf{c}$ to the stationary vector $\mathbf{c}_\infty$, that is $\Delta\mathbf{c} = \mathbf{c} - \mathbf{c}_\infty$, one obtains the temporal behavior of $\Delta\mathbf{c}$ as follows:

$$\partial_t \Delta c_i(t) = F_i(\mathbf{c}_\infty + \Delta\mathbf{c}(t)) = \sum_{j=0}^{\infty} J_{ij}(\mathbf{c}_\infty) \Delta c_j(t) + \mathcal{O}(\|\Delta\mathbf{c}\|^2), \quad \text{for } i \geq 0 , \quad (73)$$

and with Jacobian $J_{ij}(\mathbf{c}_\infty) = \frac{\partial F_i(\mathbf{c})}{\partial c_j}\big|_{\mathbf{c}=\mathbf{c}_\infty}$, whose entries are obtained after some algebra as:

$$J_{00} = -sy(p_{\text{high}} - p_{\text{low}}) , \quad (74)$$

$$J_{01} = -sy(1-y)\frac{1 - sp_{\text{high}}}{1 - sp_{\text{low}}} , \quad (75)$$

$$J_{02} = sy(1-y) , \quad (76)$$

$$J_{10} = 0 , \quad (77)$$

$$J_{11} = 0 , \quad (78)$$

$$J_{12} = 0 , \quad (79)$$

$$J_{20} = -s(1-2\lambda)(p_{\text{high}} - p_{\text{low}})^2 R'(\overline{p}_\infty) , \quad (80)$$

$$J_{21} = s(1-2\lambda)y(p_{\text{high}} - p_{\text{low}})R'(\overline{p}_\infty) , \quad (81)$$

$$J_{22} = s(1-2\lambda)(p_{\text{high}} - p_{\text{low}})((1-y)R'(\overline{p}_\infty) - 1) , \quad (82)$$

and,

$$J_{i,i+2} = -s(1-2\lambda), \quad \text{for } i \geq 1 , \quad (83)$$

$$J_{2i,2i} = -s(1-y)(p_{\text{high}} - p_{\text{low}}), \quad \text{for } i \geq 2 , \quad (84)$$

$$J_{i,j} = 0 , \quad \text{otherwise} . \quad (85)$$

The eigenvalues of the matrix $J$ determine the stability of the heterogeneous stationary distribution up to linear order in perturbations at the level of cumulants around stationarity. Its eigenvalues are given by:

- the two eigenvalues $\gamma_{1,2}$ of the $2 \times 2$ matrix,

$$\tilde{J} = \begin{pmatrix} J_{00} & J_{02} \\ J_{20} & J_{22} \end{pmatrix} \quad (86)$$

$$= \begin{pmatrix} -sy(p_{\text{high}} - p_{\text{low}}) & sy(1-y) \\ -s(1-2\lambda)(p_{\text{high}} - p_{\text{low}})^2 R'(\overline{p}_\infty) & s(1-2\lambda)(p_{\text{high}} - p_{\text{low}})((1-y)R'(\overline{p}_\infty) - 1) \end{pmatrix} , \quad (87)$$

- one eigenvalue 0,
- and infinitely many pairs of eigenvalues with values 0 and $-s(1-y)(p_{\text{high}} - p_{\text{low}}) < 0$ (because $p_{\text{high}} - p_{\text{low}} > 0$ and $1 - y > 0$ for the considered bimodal distributions).

For simplicity of the discussion, we assume $p_{\text{low}} = \min(\text{supp}(\rho_0)) = 0$ in the following, and also introduce the parameter $\beta = 2\lambda/s$ as in the main text. The two eigenvalues $\gamma_{1,2}$ of $\tilde{J}$ are given by:

$$\gamma_{1,2} = \frac{1}{2}\text{Tr}(\tilde{J}) \pm \left(\frac{1}{4}\text{Tr}(\tilde{J})^2 - \text{Det}(\tilde{J})\right)^{1/2}, \tag{88}$$

$$\text{with } \text{Tr}(\tilde{J}) = s(1 - 2\lambda)(\beta R'(\beta) - R(\beta)) - s(R(\beta) - \beta), \tag{89}$$

$$\text{and } \text{Det}(\tilde{J}) = s^2(1 - 2\lambda)R(\beta)(R(\beta) - \beta). \tag{90}$$

## Linear stability for small $\lambda$

Under the assumptions $R(0) = 0$ and $1 < R'(0) < \infty$, one checks that for $0 < \lambda \ll 1$ the eigenvalues of the Jacobian $\tilde{J}$ in **Equation (88)** are given by:

$$\gamma_{1,2} = -\lambda(R'(0) - 1) + \mathcal{O}(\lambda^3) \pm i\lambda\left((R'(0) - 1)(3R'(0) + 1) + \mathcal{O}(\lambda)\right)^{1/2}, \tag{91}$$

and, thus, $\text{Re}(\gamma_{1,2}) < 0$ as $\lambda \searrow 0$.

Therefore, for small response probabilities, the heterogeneous stationary distribution (**Equation (2)** of the main text) is stable up to linear order in perturbations at the level of cumulants around stationarity (here shown under the assumptions $p_{\text{low}} = \min(\text{supp}(\rho_0)) = 0$, $R(0) = 0$, and $1 < R'(0) < \infty$).

## Linear stability for the response function $R(\beta) = \beta + \kappa \cdot \sin(\pi\beta)$

Upon choosing the response function $R(\beta) = \beta + \kappa \cdot \sin(\pi\beta)$ with $\beta \in [0,1]$ (that is $\lambda \in [0, s/2]$) and with $\kappa \in [0, 1/\pi]$, one checks that all eigenvalues of the Jacobian $\tilde{J}$ in **Equation (88)** have negative real part.

Therefore, for the special choice of the response function that up-regulates the cellular autoinducer production for all sensed average productions in the population, all heterogeneous stationary distributions (**Equation (2)** of the main text) for choices of the parameters $\lambda \in [0, s/2]$ and $\kappa \in [0, 1/\pi]$ are stable up to linear order in perturbations at the level of cumulants around stationarity.

