## [Decision Letter]

Thank you for submitting your article "Eco-evolutionary dynamics in quorum-sensing microbial populations can induce heterogeneous production of autoinducers" for consideration by *eLife*. Your article has been favorably evaluated by Gisela Storz (Senior Editor) and three reviewers, one of whom, is a member of our Board of Reviewing Editors. The reviewers have opted to remain anonymous.

The reviewers have discussed the reviews with one another and the Reviewing Editor has drafted this decision to help you prepare a revised submission.

Summary:

The paper presents a theoretical model for the emergence of phenotypic heterogeneity in autoinducer production, which is essential for quorum sensing in bacteria. The paper shows that a feedback between the total concentration of autoinducers and individual costly production of autoinducers is sufficient to generate heterogeneity in individual production.

Essential revisions:

The reviewers found the paper very interesting and well written. However, they raised a number of concerns that preclude publication of the paper in its present form.

Notably, some comments point to a lack of integration of the theoretical results with empirical reality. On the one hand, the paper should make more specific suggestions about how the proposed theory can be tested. On the other hand, and more importantly, the authors should consider incorporating more biological realism into their modelling. Reviewer 3 makes a number of useful suggestions in that regard. For example, this reviewer suggests incorporating heterogeneity in signal perception, an issue that was also commented on by reviewers 1 and 2. Further, it is unclear how decay of the signal concentration in the environment would affect the results. You will see that the reviewers have made a number of other suggestions about how to improve the paper.

If you find that you are able to address the concerns raised in the reviewer comments we would welcome a resubmission of your article. In the revision, please try to address all of the reviewers' comments in a constructive way. In particular, it would be good if you could extend the theoretical analysis to cover the salient comments.

*Reviewer #1:*

This is an interesting paper showing how the emergence of phenotypic heterogeneity in autoinducer production can, in principle, be understood as an emergent collective behaviour of quorum sensing in microbial ecosystems.

The model presented is simple and elegant, and results are supported by both numerical simulations and analytical derivations. The main result of long transients at heterogeneous states in the stochastic model is surprising, and the mean-field approximation serves the purpose of explaining this result very well.

In principle, the analytical work generates novel and testable predictions for autoinducer heterogeneity in quorum sensing (although the paper might benefit from being more specific in this regard).

1) The phenotypic response defined by the response function R is assumed to be deterministic (as e.g. seen by the Delta function appearing in the autoinducer equation (1)). I think it is important to investigate what happens if the response is instead probabilistic, i.e., when the offspring phenotype is drawn probabilistically from some distribution with mean R() and positive variance.

2) I don't agree with the paper's distinction between "ecological" and "evolutionary" dynamics. The authors call the feedback mechanism for autoinducer production "ecological", and they call differences in growth rates "evolutionary". However, there are no genetic differences between individuals, and hence no evolutionary dynamics occur. Instead, individuals simply have different birth rates, which is a purely *ecological* difference. Thus, both processes, fitness difference and global feedback, are clearly ecological in this model, and any claim that the former is "evolutionary" is misleading. The coupling is not between "ecological" and "evolutionary" dynamics, but between global production and individual birth rates.

*Reviewer #2:*

This manuscript shows how phenotypic heterogeneity in autoinducer (AI) production may arise in monostable autoregulation from the interplay between sensing and responding to the environment and fitness differences between producer and non-producer phenotypes. I really enjoyed the manuscript, and feel that this is an important contribution to the study of collective behavior in microbes, mainly because it provides an alternative mechanism to explain AI production phenotypic heterogeneity using "bistable threshold models". The paper is well-written and technically very rigorous. In my view, it would be acceptable for publication in *eLife* after some minor points are clarified.

According to this model, phenotypic heterogeneity relies on (i) fitness differences between producers and non-producers linked to the metabolic cost associated with AI production, and (ii) AI production, p, is more likely to be inherited from the parental cell than obtained from the mean level of production, , in the environment. I am wondering if, due to this fitness difference between producers and nonproducers the authors could devise any population level experimental procedure (in addition to the single cell experiments briefly discussed in the last paragraph of the Discussion) to check the existence of the proposed feedback. The authors envision this possibility, but I think that the paper would greatly improve by establishing tighter connections between theoretical results and their possible empirical confirmation. In my opinion, this would be an important point to reach the broad readership of *eLife*, although the theoretical results are of enough significance by their own.

I have some doubts about the probabilistic mechanism by which newborn cells adopt. In my opinion, this is the most important ingredient of the model, because it allows for phenotypic heterogeneity, and I think that it would be good to discuss whether there is empirical support for such election (is the production level a trait maintained during the whole life cycle of the cell?) or whether that is a choice of the authors. In the latter case, it could be interesting to discuss other possible mechanisms and maybe outline the robustness of the results presented here against these alternatives.

I think that the importance of the paper could be highlighted, especially for non-specialist readers, if the Introduction is slightly reorganized. As it is now, one may understand that there are experimental proofs of both bistable and monostable autoinducer synthesis regulation, and that the authors provide a mechanism that could explain phenotypic heterogeneity in the latter case. However, if I understood the Discussion correctly (second paragraph), bistable autoregulation has not been experimentally verified, but there are not empirical studies showing monostable autoregulation either. I think this point should appear earlier in the Introduction, accompanied by more references to existent models that utilize a bistable autoregulation (1,2). My feeling is that if there is not experimental verification of monostable regulation in AI synthesis, then this paper opens a much broader and deeper question because it not only provides a new mechanism by which phenotypic heterogeneity can emerge, but also suggests that AI synthesis regulation could be monostable. In either case, I think that this should be clarified.

1) Goryachev AB, Toh DJ, Wee KB, Lee T, Zhang HB, et al. (2005) Transition to Quorum Sensing in an Agrobacterium Population: A Stochastic Model. PLOS Computational Biology 1(4): e37. doi: 10.1371/journal.pcbi.0010037

2) Dockery, Jack D., and James P. Keener. "A mathematical model for quorum sensing in *Pseudomonas* aeruginosa." Bulletin of mathematical biology 63.1 (2001): 95.

*Reviewer #3:*

The present study describes theoretical model to provide a possible explanation for the occurrence of phenotypic heterogeneity in quorum sensing. In general phenotypic heterogeneity is an interesting topic which spans from antibiotics persistence to bacterial competence. Resent study has shown that bacteria exhibit reversible heterogeneity in QS response even in the presence of saturating concentration of QS signal, which indicate that bacteria maintain a stochastic homogenous population as a bet-hedging strategy to counter fluctuating environmental conditions.

1) The model has been thought in the lines of heterogeneous expression of QS synthase gene by epigenetic mechanism, however in QS, sensor plays an important role in the perception and auto regulation of the QS synthase as well as expression of other QS controlled genes. In this model however, the heterogeneity in terms of QS signal perception is not considered which could play an important role in the process.

2) The model is based on the assumption that there appears to be little change in the extracellular signal concentration once it is produced or the intracellular conc. It has now been shown that signal degradation also takes place in some QS system such as in *Pseudomonas* syringae, in which bacteria secrete QS degrading enzymes (some are secreted and some are intracellular), which can also influence the heterogeneity in the population. However, it appears that in this model it could have been also considered while simulating the fluctuation along with the assumption of non-producers grows faster.

3) It has been shown that in certain environmental condition, the QS nonproducers has a growth advantage as they take benefit of the social task performed by the responders and also save on signal production cost. However, it has also has been shown in case of *Pseudomonas* that QS- strains has a big disadvantage under certain environmental condition where QS controlled "private goods" are required (Science. 2012 Oct 12;338(6104):264-6.). How will this affect the present model?

4) How the stability of auto inducers in the system will influence the cellular response to fluctuating environment and phase transition to homogeneous response?

5) A factor that could play a role in the QS heterogeneity response is the relative affinity of different QS signal with the receptor. In case the affinity is high, the local concentration of QS signal could be maintained for a prolong period and can affect the distribution. In this model, it is better to include the signal perception component also, as ultimately even for the stochastic expression of signal synthase, the signal perception and henceforth regulation of gene expression also contributes to the overall QS process.

6) What happens when competition experiments are performed with signal blind and signal sensing mutants in this model? Does fluctuating the ratio of these variants also influence the heterogeneous distribution?

---

## [Author Response]

*Essential revisions:*

*The reviewers found the paper very interesting and well written. However, they raised a number of concerns that preclude publication of the paper in its present form.*

*Notably, some comments point to a lack of integration of the theoretical results with empirical reality.*

We thank you and reviewers #1, #2, and #3 for the careful reading of our manuscript and for the pertinent comments. Following your suggestions, we have made changes to our manuscript in order to integrate our theoretical results with empirical reality. In particular, we greatly appreciate the suggestion to extend the Discussion of our results in the light of experimental microbiology with respect to model assumptions, model predictions, and their robustness. The following list summarizes the essential changes to the manuscript. Further details are provided in the respective point-by-point response to the reviewer’s comments; see below.

1) We significantly extended the discussion of our results in the “Discussion” section with focus on integrating our theoretical results with empirical reality. We discuss in detail our model assumptions and added experimental studies to the text that support our assumptions. We also indicate experimental directions to verify or falsify all other model assumptions. Furthermore, we discuss the experimental implications of our results for phenotypic heterogeneity in quorum-sensing microbial populations and propose directions for future experiments to test these implications. We included the following points to the manuscript:

1.1) Subsection “Does autoinducer production reduce individual growth rate?”. Discussion of the correlation between the cellular production degree of autoinducers and an individual’s growth rate.

We now describe in detail the experimental background of our model assumption that producers of large autoinducer molecules reproduce slower than non-producers.

We include the recent experimental study of [Ruparell et al., 2016] (DOI: 10.1038/srep33101) into our discussion, in which metabolic costs were quantified in terms of reproduction rates for a microbial strain that uses AHL molecules as autoinducers for signaling. This study carefully quantifies the consequences of signaling on a strain’s fitness by conducting experiments in both mono and mixed culture with signal blind and signal sensing mutants as envisaged by reviewer #3.

Furthermore, we include the study of [He et al., 2003] (DOI: 10.1128/JB.185.3.809-822.2003) into the “Discussion” section. This study showed growth impairments due to signaling for the strain *Sinorhizobium fredii* NGR234 (the same strain for which phenotypic heterogeneity heterogeneity in the expression of autoinducer synthase genes was observed in [Grote et al., 2014]). Therefore, it would be interesting to explore experimentally whether the phenotypic heterogeneity observed in *Sinorhizobium fredii* NGR234 could be explained through the eco-evolutionary mechanism proposed by our quorum-sensing model.

We also rearranged the model definition in the section “Set-up of the quorum-sensing model” in that the theoretical estimates of [Keller and Surette, 2006] for the metabolic costs due to signaling are now shifted to the “Discussion” section. This way, the model definition becomes more streamlined as suggested by reviewer #2.

1.2) Subsection “A question of spatio-temporal scales: How stable and how dispersed are autoinducers in the environment?”. Discussion of the spatio-temporal scales determining the stability and the dispersal of autoinducers, and underlying the assumptions in our quorum-sensing model.

We assume in the quorum-sensing model that autoinducers are uniformly degraded in a well-mixed environment. These assumptions do not hold true for a spatially structured microbial biofilm, but should be fulfilled during the stationary phase of microbial growth in a well-mixed batch culture. In particular, we now refer to the experimental studies of [Yates et al., 2002] (DOI: 10.1128/IAI.70.10.5635-5646.2002) and [Byers et al., 2002] (DOI: 10.1128/jb.184.4.1163-1171.2002), in which pH and temperature-dependent degradation of autoinducers in the environment are discussed (for example, the half-life time of AHL molecules can be less than 30 min at pH = 8.5 and temperature = 37 degree Celsius; this time scale is, thus, much shorter than typical microbial doubling times). These studies support our model assumptions of a high degradation rate of autoinducers in the environment under conditions that are fulfilled, for example, during stationary phase of microbial growth.

1.3) Subsection “How is production of autoinducers up-regulated at the single-cell level?”. Discussion of the regulation of autoinducer production at the single-cell level.

As proposed by the comments of reviewers #2 and #3, we now discuss in greater detail the regulation of autoinducer production at the cellular level. In particular, we detail on the difference between monostable and bistable up-regulation of autoinducer synthesis. Our analysis shows that the qualitative form of the regulation could discriminate between different mechanisms that control phenotypic heterogeneity of the autoinducer production at the population level. We describe how phenotypic heterogeneity may arise through stochastic gene expression for bistable gene regulation (but not for monostable regulation). We then contrast this mechanism with the eco-evolutionary mechanism identified in our study, which induces phenotypic heterogeneity for both monostable and for bistable regulation.

As envisaged by reviewer #2, we now mention that observations at the population level of both phenotypic heterogeneity in autoinducer synthesis and of growth rate differences between producers and non-producers could be an indicator for monostable regulation of autoinducer synthesis at the cellular level.

1.4) Subsection “How is production of autoinducers up-regulated at the single-cell level?”, last paragraph. Discussion of timescales on which microbes respond to autoinducers in the environment.

In the implementation of our quorum-sensing model, individuals respond to the environment stochastically at reproduction events. Reviewer #2 asked for a discussion of this implementation. The rule that offspring individuals can only respond at reproduction events represents a coarse-grained view in time to facilitate the mathematical analysis and to identify the eco-evolutionary mechanism by which phenotypic heterogeneity can be induced. The response probability can actually be interpreted as the rate with which individuals respond to autoinducers in the environment. This cellular response rate is then effectively measured in units of the cell’s reproduction rate in the quorum-sensing model. This can also be inferred from the mean-field equation and the prefactor 2\lambda of the sense-and-response term. Even though sense and response to the environment are implemented at reproduction events in the model, effective changes of the population due to sense-and-response occurs at rate 2\lambda in the macroscopic description.

To quantify such response rates in a microbiological setting, experiments at the single-cell level seem most promising to us at present (see point 1.5 below). We hope that our work stimulates such experimental studies.

1.5) Subsection “Single-cell experiments”. Discussion of single-cell experiments.

We now discuss why and how single-cell experiments would help to verify or falsify our model assumptions (for example, to determine the form of the fitness and *response* function, to quantify the selection strength and response probability, and to refine the model set-up).

More explicitly, it would be desirable to simultaneously monitor, at the single-cell level, the correlations between autoinducer levels in the environment, the expression of autoinducer synthase genes, and the transcriptional regulators that mediate response to quorum sensing. Upon adjusting the level of autoinducers in a controlled manner, for example in a microfluidic device, one could characterize how cells respond to autoinducers in the environment.

This way, it might be possible to answer questions of:

i) how the cellular production of autoinducers is regulated (monostable or bistable regulation, or a different form of regulation), ii) whether response times to environmental changes are stochastic and whether response rates can be identified, iii) as to what extent cellular response in the production of autoinducers depends on both the level of autoinducers in the environment and on the cell’s present production degree, and (iv) how production of autoinducers is correlated with single-cell growth rate.

1.6) Subsection “What is the function of phenotypic heterogeneity in autoinducer production?”. Outlook for experiments and theory: function of phenotypic heterogeneity in the autoinducer production.

As an outlook for future experimental and theoretical work, and also in response to reviewer #1 and #3, we now explicitly mention that our work focuses on how phenotypic heterogeneity in the autoinducer production may be controlled, but that our work does not address the function of this phenomenon. The evolutionary contexts and ecological scenarios under which phenotypic heterogeneity in the autoinducer production may have evolved are still to be investigated from an experimental point of view (see [Grote et al., 2014], [Garmyn et al., 2011]). From a modeling perspective, it may be possible to extend, for example, our chosen fitness function with a term that explicitly accounts for the benefit of signaling either at the cellular or population level, and study suitable evolutionary contexts and possible ecological scenarios.

2) We have extensively tested that the observed phenotypic heterogeneity is robust against incorporating more biological realism into our model set-up, as suggested by you and all reviewers. More specifically, we have checked both numerically and mathematically that bimodal quasi-stationary states still arise in the relevant parameter regimes when noisy inheritance of the production degree, noisy perception of the average production level, and noisy response to it are included into the model set-up. Noise was always taken as Gaussian noise with zero mean and positive variance. These results show that the eco-evolutionary mechanism for phenotypic heterogeneity is not a fine-tuned effect, but may arise as well if more biochemical details of quorum sensing are accounted for from a theoretical point of view. Therefore, we expect the eco-evolutionary mechanism to be relevant for microbial experiments.

Corresponding changes to the manuscript are made as follows:

· Representative trajectories of the quorum-sensing model including noise at the above-mentioned levels (inheritance, perception, response) are now included in the manuscript as Appendix Figure 2. All details of the numerical implementation are provided in the Figure caption.

· We integrated these new numerical simulations with our mathematical analysis by connecting the numerically observed robustness of the quasi-stationary states with the linear stability analysis of heterogeneous bimodal distributions of the quorum-sensing model [Subsection “Results of mathematical analysis”, seventh paragraph].

· We now explicitly mention the robustness of phenotypic heterogeneity through the eco-evolutionary mechanism throughout the text: at the end of the subsection “Set-up of the quorum-sensing model”, in the subsection “Results of numerical simulations”, fourth paragraph, in the subsection “Results of mathematical analysis”, seventh and last paragraphs.

3) We improved the wording and corrected typos throughout the manuscript.

*On the one hand, the paper should make more specific suggestions about how the proposed theory can be tested.*

We are happy to implement this proposal. Point 1 in the response above provides all details to experimentally test our theory and lists all according changes to the manuscript.

*On the other hand, and more importantly, the authors should consider incorporating more biological realism into their modelling. Reviewer 3 makes a number of useful suggestions in that regard. For example, this reviewer suggests incorporating heterogeneity in signal perception, an issue that was also commented on by reviewers 1 and 2.*

We appreciate this comment and have carried out extensive numerical simulations to test the robustness of the results of our theory against incorporating more biological realism into the set-up of the quorum-sensing model (noisy inheritance, noisy perception, and noisy response). Point 2 in the response above provides all details and lists the according changes to the manuscript.

*Further, it is unclear how decay of the signal concentration in the environment would affect the results.*

We are thankful for this comment because it led us to formulate more explicitly the involved time scales underlying our theoretical model. Time scales that determine the stability of autoinducers in the environment are discussed in point 1.2 above; according changes to the manuscript are implemented in a separate paragraph [subsection “A question of spatio-temporal scales: How stable and how dispersed are autoinducers in the environment?”].

*You will see that the reviewers have made a number of other suggestions about how to improve the paper.*

We thank the reviewers for their detailed reading and the helpful comments. Our responses to their specific comments and suggestions, and the accompanied changes to the manuscript are attached below.

*If you find that you are able to address the concerns raised in the reviewer comments we would welcome a resubmission of your article. In the revision, please try to address all of the reviewers' comments in a constructive way. In particular, it would be good if you could extend the theoretical analysis to cover the salient comments.*

*Reviewer #1:*

*[…] In principle, the analytical work generates novel and testable predictions for autoinducer heterogeneity in quorum sensing (although the paper might benefit from being more specific in this regard).*

*1) The phenotypic response defined by the response function R is assumed to be deterministic (as e.g. seen by the Delta function appearing in the autoinducer equation (1)). I think it is important to investigate what happens if the response is instead probabilistic, i.e., when the offspring phenotype is drawn probabilistically from some distribution with mean R() and positive variance.*

We agree with the reviewer that it is an interesting question to ask how noisy response affects phenotypic heterogeneity in our quorum-sensing model. We now demonstrate in detail that phenotypic heterogeneity in our quorum-sensing model is robust against noise at the level of inheritance of the production degree, perception of the average production level, and response; see our detailed response 2 in the answer to the Reviewing Editor.

*2) I don't agree with the paper's distinction between "ecological" and "evolutionary" dynamics. The authors call the feedback mechanism for autoinducer production "ecological", and they call differences in growth rates "evolutionary". However, there are no genetic differences between individuals, and hence no evolutionary dynamics occur. Instead, individuals simply have different birth rates, which is a purely ecological difference. Thus, both processes, fitness difference and global feedback, are clearly ecological in this model, and any claim that the former is "evolutionary" is misleading. The coupling is not between "ecological" and "evolutionary" dynamics, but between global production and individual birth rates.*

We agree with the reviewer that the term “evolutionary dynamics” is used to mean different phenomena in different fields of (theoretical) biology. “Evolutionary dynamics” is used both in the field of population genetics (to term changes of genotypes) and in the field of population dynamics (to term changes of phenotypes). From a theoretical point of view, it is ultimately a question of time scales whether one terms the *temporal change of a dynamic variable* attached to an individual in a population as “evolutionary” or not. Since our manuscript deals with phenotypic heterogeneity (that is, the coexistence of several phenotypes in a population of genetically identical individuals), we use the vocabulary that is common in the field of phenotypic heterogeneity; see, for example, the text books of [Maynard-Smith, Evolution and the Theory of Games, 1982], [Hofbauer and Sigmund, Evolutionary Games and Population Dynamics, 1998], and [Nowak, Evolutionary Dynamics, 2006].

In order to make clearer that the context of “evolutionary” is meant in our work at the level of phenotypes in a population of individuals, we changed the wording slightly at the following points:

· “The key aspect of our work is how [new: the composition of] a population evolves in time when its constituents respond to an environment that is being shaped by their own activities.”

· “The coupling of ecological dynamics (given by the average production level of autoinducers ) with evolutionary dynamics (determined by fitness differences [new: between the phenotypes]) through quorum sensing results in interesting collective behavior, as we show next.”

· See also the changes to the manuscript in response to the next comment.

*Reviewer #2:*

*[…] According to this model, phenotypic heterogeneity relies on (i) fitness differences between producers and non-producers linked to the metabolic cost associated with AI production, and (ii) AI production, p, is more likely to be inherited from the parental cell than obtained from the mean level of production, , in the environment. I am wondering if, due to this fitness difference between producers and nonproducers the authors could devise any population level experimental procedure (in addition to the single cell experiments briefly discussed in the last paragraph of the Discussion) to check the existence of the proposed feedback. The authors envision this possibility, but I think that the paper would greatly improve by establishing tighter connections between theoretical results and their possible empirical confirmation. In my opinion, this would be an important point to reach the broad readership of eLife, although the theoretical results are of enough significance by their own.*

We thank the reviewer for the comments and the detailed suggestions. As described in detail in response 1 of the answer to the Reviewing Editor, we now make a couple of suggestions as to how our model assumptions are supported by experimental findings or how they might be verified or falsified from an experimental point of view. We now discuss in the manuscript:

· the correlation between the cellular production degree of autoinducers and an individual’s growth rate [subsection “Does autoinducer production reduce individual growth rate?”],

· the spatio-temporal scales that determine the stability and the dispersal of autoinducers and that are assumed in our model [subsection “A question of spatio-temporal scales: How stable and how dispersed are autoinducers in the environment?”],

· the regulation of autoinducer production at the single-cell level [subsection “How is production of autoinducers up-regulated at the single-cell level?”],

· the timescales on which microbes respond to autoinducers in the environment [subsection “How is production of autoinducers up-regulated at the single-cell level?”, last paragraph],

· single-cell experiments [subsection “Single-cell experiments”], and

· an outlook for experiments and theory regarding the function of phenotypic heterogeneity in the autoinducer production [subsection “What is the function of phenotypic heterogeneity in autoinducer production?”].

*I have some doubts about the probabilistic mechanism by which newborn cells adopt. In my opinion, this is the most important ingredient of the model, because it allows for phenotypic heterogeneity, and I think that it would be good to discuss whether there is empirical support for such election (is the production level a trait maintained during the whole life cycle of the cell?) or whether that is a choice of the authors. In the latter case, it could be interesting to discuss other possible mechanisms and maybe outline the robustness of the results presented here against these alternatives.*

We fully agree with the reviewer that our rule how cells adopt is a simplification of the reality. The rule that offspring individuals can only respond at reproduction events is a coarse-grained view in time to facilitate the mathematical analysis and to identify the eco-evolutionary mechanism. The response probability can actually be interpreted as the rate with which individuals respond to autoinducers in the environment; see response 1.4 of the answer to the Reviewing Editor. We now discuss this assumption in detail in the Discussion section [subsection “How is production of autoinducers up-regulated at the single-cell level?”, last paragraph].

The robustness of the quorum-sensing model is supported by new numerical results, by our mean-field theory, and by the linear stability analysis of heterogeneous, quasi-stationary states of the population; please see response 2 of the answer to the Reviewing Editor.

*I think that the importance of the paper could be highlighted, especially for non-specialist readers, if the Introduction is slightly reorganized. As it is now, one may understand that there are experimental proofs of both bistable and monostable autoinducer synthesis regulation, and that the authors provide a mechanism that could explain phenotypic heterogeneity in the latter case. However, if I understood the Discussion correctly (second paragraph), bistable autoregulation has not been experimentally verified, but there are not empirical studies showing monostable autoregulation either. I think this point should appear earlier in the Introduction, accompanied by more references to existent models that utilize a bistable autoregulation (1,2). My feeling is that if there is not experimental verification of monostable regulation in AI synthesis, then this paper opens a much broader and deeper question because it not only provides a new mechanism by which phenotypic heterogeneity can emerge, but also suggests that AI synthesis regulation could be monostable. In either case, I think that this should be clarified.*

*1) Goryachev AB, Toh DJ, Wee KB, Lee T, Zhang HB, et al. (2005) Transition to Quorum Sensing in an Agrobacterium Population: A Stochastic Model. PLOS Computational Biology 1(4): e37. doi: 10.1371/journal.pcbi.0010037*

*2) Dockery, Jack D., and James P. Keener. "A mathematical model for quorum sensing in Pseudomonas aeruginosa." Bulletin of mathematical biology 63.1 (2001): 95.*

We thank the reviewer for the interesting comment. Accordingly, we made the following changes to the manuscript:

· We now mention explicitly in the Introduction: “From an experimental point of view it is often not known, however, whether autoinducer synthesis is up-regulated for all autoinducer levels or only above a threshold level.”

· We added the proposed references (Goryachev et al., 2005; Dockery and Keener 2001) to the manuscript. We thank the reviewer for pointing us to these early mathematical models of quorum-sensing.

· We now discuss extensively the regulation of autoinducer production at the single-cell level in the subsection “How is production of autoinducers up-regulated at the single-cell level?”, and have taken up many points suggested by the reviewer; see also response 1.3 of the answer to the Reviewing Editor for details.

*Reviewer #3:*

*[…] 1) The model has been thought in the lines of heterogeneous expression of QS synthase gene by epigenetic mechanism, however in QS, sensor plays an important role in the perception and auto regulation of the QS synthase as well as expression of other QS controlled genes. In this model however, the heterogeneity in terms of QS signal perception is not considered which could play an important role in the process.*

We agree with the reviewer that it is an interesting question to ask how noisy response affects phenotypic heterogeneity in our quorum-sensing model. We now demonstrate in detail that phenotypic heterogeneity in our quorum-sensing model is robust against noise at the level of inheritance of the production degree, perception of the average production level, and response; see our detailed response 2 in the answer to the Reviewing Editor.

*2) The model is based on the assumption that there appears to be little change in the extracellular signal concentration once it is produced or the intracellular conc. It has now been shown that signal degradation also takes place in some QS system such as in Pseudomonas syringae, in which bacteria secrete QS degrading enzymes (some are secreted and some are intracellular), which can also influence the heterogeneity in the population. However, it appears that in this model it could have been also considered while simulating the fluctuation along with the assumption of non-producers grows faster.*

We are thankful for this comment because it led us to formulate more explicitly the involved time scales underlying our theoretical model. Time scales that determine the stability of autoinducers in the environment are discussed in response 1.2 of the answer to the Reviewing Editor; we discuss signal degradation in the environment now in a separate paragraph [subsection “A question of spatio-temporal scales: How stable and how dispersed are autoinducers in the environment?”]. In essence, we assume in the quorum-sensing model that autoinducers are uniformly degraded at a high rate in a well-mixed environment. A high rate refers here to a time scale that needs to be compared to the strain’s doubling time (for example, the half-life time of AHL molecules can be less than 30 min at pH = 8.5 and temperature = 37 degree Celsius; this time scale is, thus, much shorter than typical microbial doubling times). Our assumptions of high autoinducer degradation in the environment may not hold true for a spatially structured microbial biofilm, but may be fulfilled during the stationary phase of microbial growth in a well-mixed batch culture.

*3) It has been shown that in certain environmental condition, the QS nonproducers has a growth advantage as they take benefit of the social task performed by the responders and also save on signal production cost. However, it has also has been shown in case of Pseudomonas that QS- strains has a big disadvantage under certain environmental condition where QS controlled "private goods" are required (Science. 2012 Oct 12;338(6104):264-6.). How will this affect the present model?*

The reviewer raises two interesting points with this comment that we are happy to discuss now in the “Discussion” section.

· How does the spatial dispersal of autoinducers in the environment affect the observed phenotypic heterogeneity? We now discuss the scales that determine the dispersal of autoinducers and that are assumed in our model [subsection “A question of spatio-temporal scales: How stable and how dispersed are autoinducers in the environment?”]. Spatial dispersal of autoinducers in the population depends, for example, upon cellular mechanisms that import and export autoinducers into the cell from the environment and vice versa, and upon the spatial structure of the microbial population. The degree of dispersal determines whether autoinducers remain spatially privatized to a single cell, diffuse to neighboring cells, or are spread evenly between all cells of the population. Consequently, the spatial organization of the microbial population determines as to what extent microbes sense rather the global or a local average production level.

· How do benefits of signaling (either to the cellular level or at the population level) affect the observed phenotypic heterogeneity? From a modeling perspective, this question points in the direction of the function of phenotypic heterogeneity in the autoinducer production, which we briefly discuss now as an outlook for experiments and theory [subsection “What is the function of phenotypic heterogeneity in autoinducer production?”]. It might be possible, for example, to extend our chosen fitness function with a term that explicitly accounts for the benefit of signaling either at the cellular or population level, and study suitable evolutionary contexts and possible ecological scenarios; also privatization of the signal as a public good may be implemented as the reviewer suggests. Heuristically speaking, it seems straightforward from a theoretical point of view that producing traits may proliferate more easily in (quorum-sensing) microbial populations if signal molecules confer an explicit benefit to single cells and if signal molecules remain privatized. We have the impression that modeling the ecological and evolutionary contexts under which phenotypic heterogeneity in the autoinducer production might have evolved would go beyond the scope of the current manuscript. Furthermore, these points are still under investigation from an experimental point of view (see [Grote et al., 2014], [Garmyn et al., 2011]). Therefore, we formulated these thoughts as an outlook in the aforementioned subsection.

*4) How the stability of auto inducers in the system will influence the cellular response to fluctuating environment and phase transition to homogeneous response?*

Again, this is a very interesting question from an experimental point of view. In our model, we assume that autoinducers are degraded at a rate that is higher than the cellular growth rate. We now discuss ecological scenarios under which this assumption may hold true and conclude that the stationary phase of microbial growth in batch culture might provide such a scenario. Time scales that determine the stability of autoinducers in the environment are further explained in response 1.2 of the answer to the Reviewing Editor; we discuss the stability of autoinducers in the environment now in a separate paragraph of the Discussion section [subsection “A question of spatio-temporal scales: How stable and how dispersed are autoinducers in the environment?”].

*5) A factor that could play a role in the QS heterogeneity response is the relative affinity of different QS signal with the receptor. In case the affinity is high, the local concentration of QS signal could be maintained for a prolong period and can affect the distribution. In this model, it is better to include the signal perception component also, as ultimately even for the stochastic expression of signal synthase, the signal perception and henceforth regulation of gene expression also contributes to the overall QS process.*

We agree with the reviewer that it would be possible to include further molecular details of quorum sensing into our model. At present, we capture all molecular and biochemical details by the response function on a macroscopic level. The response function encapsulates all steps involved in the autoinducer production between perception of the average production level (“input” of signal) and adjustment of the individual production degree in response (“output” of producing the signal). Because many different quorum-sensing architectures are known (see, for example, [Waters and Bassler, 2005]) and relevant to our work, a specific choice of regulation architecture will narrow the scope of the results of our study. Therefore, we decided to capture all steps between signal perception and regulation by one effective function, which we introduced as the response function. Such response functions are actually measured, as for example depicted in Figure 3 of [He et al., 2003] (DOI: 10.1128/JB.185.3.809-822.2003; termed dose-response curve for an acyl-HSL as autoinducer). This way, our model remains more flexible and adjustable to specific experimental situations if one is interested in fitting a model to experimental data. On the other hand, the response function captures the necessary condition of up-regulation to explain the mechanism of phenotypic heterogeneity in autoinducer production through the eco-evolutionary mechanism.

*6) What happens when competition experiments are performed with signal blind and signal sensing mutants in this model? Does fluctuating the ratio of these variants also influence the heterogeneous distribution?*

We thank the reviewer for this comment.

· First, this question anticipates exactly the experiments that we envision, for example, to determine the correlation between the cellular production degree of autoinducers and an individual’s growth rate. We refer the reviewer to point 1.1 in the response to the Reviewing Editor and according changes to the manuscript [subsection “Does autoinducer production reduce individual growth rate?”].

In brief, we now added the study of [Ruparell et al., 2016] (DOI: 10.1038/srep33101), which showed a reduced fitness of a producing strain (producing a long-chain AHL; OC$_{12}$-HSL, synthesized via the las operon) in both mono and mixed culture compared with a non-producing strain. It would be interesting to conduct similar experiments for the strain *Sinorhizobium fredii* NGR234 for which a heterogeneous synthesis of autoinducers was reported.

· Second, we have checked extensively the robustness of phenotypic heterogeneity in our model. We refer the reviewer to point 2 in the response to the Reviewing Editor for a detailed answer and the according list of changes to the manuscript. In essence, perturbations of the ratio between producers and non-producers in the heterogeneous state of the population decay in our quorum-sensing model; phenotypic heterogeneity through the eco-evolutionary mechanism is a robust outcome.